# AttributionLab: Faithfulness of Feature Attribution Under Controllable Environments

## Abstract

Feature attribution explains neural network outputs by identifying relevant input features. How do we know if the identified features are indeed relevant to the network? This notion is referred to as *faithfulness*, an essential property that reflects the alignment between the identified (attributed) features and the features used by the model. One recent trend to test faithfulness is to design the data such that we know which input features are relevant to the label and then train a model on the designed data. Subsequently, the identified features are evaluated by comparing them with these designed ground truth features. However, this idea has the underlying assumption that the neural network learns to use *all* and *only* these designed features, while there is no guarantee that the learning process trains the network in this way. In this paper, we solve this missing link by *explicitly designing the neural network* by manually setting its weights, along with *designing data*, so we know precisely which input features in the dataset are relevant to the designed network. Thus, we can test faithfulness in *AttributionLab*, our designed synthetic environment, which serves as a sanity check and is effective in filtering out attribution methods. If an attribution method is not faithful in a simple controlled environment, it can be unreliable in more complex scenarios. Furthermore, the AttributionLab environment serves as a laboratory for controlled experiments through which we can study feature attribution methods, identify issues, and suggest potential improvements.

## 1 Introduction

Neural networks exhibit increasing capabilities as the scale of their design and their training data increases (Bubeck et al., 2023; Wei et al., 2022; Oquab et al., 2023; OpenAI, 2023; Caron et al., 2021; Brown et al., 2020). These capabilities are achieved through the use of basic architectural blocks (Simonyan & Zisserman, 2015; He et al., 2016; Vaswani et al., 2017). Though we know the architectural design of these networks and know their computational graph explicitly, we do not have a human interpretable understanding of neural networks. One way to explain the neural network output is to *identify important input features* for a single prediction, an explanation paradigm known as *input feature attribution*. There has been an ongoing quest for finding attribution methods to explain neural network functions (Selvaraju et al., 2017; Zeiler & Fergus, 2014; Bach et al., 2015; Springenberg et al., 2015; Shrikumar et al., 2017; Fong et al., 2019; Lundberg & Lee, 2017; Zhang et al., 2021; Sundararajan et al., 2017; Schulz et al., 2020). However, one challenge remains: How can we know whether the features identified by attribution are aligned with features relevant to the neural network? How do we know if an attribution is *faithful*? An attribution may seem reasonable to us, but the neural network may use other input features (Ilyas et al., 2019). Conversely, an attribution may seem unreasonable but be faithful and indeed reflect the features relevant to the neural networks. Moreover, in the presence of multiple differing attribution explanations (Krishna et al., 2022; Khakzar et al., 2022), it is unclear which explanation to trust. The growing complexity of both networks and feature attribution methodologies adds further intricacy to the problem.

One recent trend to assess the faithfulness of attribution methods is through the use of synthetic data. Synthetic data are designed such that associations between features and labels are known to users (Arras et al., 2022; Agarwal et al., 2023; Zhou et al., 2022). *However, as we discuss in Section 3 there is no guarantee that the learning process will train the network to use the designed associations in the dataset.* Hence, the association learned by models can differ from the designed

association in the synthetic dataset. Consequently, evaluation based on the designed association is not guaranteed to reflect the faithfulness of feature attribution methods. Furthermore, many evaluations usually report a performance score without any information on why an attribution method has limited performance. A test environment capable of uncovering properties and issues of methods can better contribute to the development of feature attribution research.

In scientific experiments with complex setups and multiple variables, it is typical to use a laboratory setting to conduct controlled tests. Analogously, our work proposes a paradigm for providing a controlled laboratory environment. In this laboratory environment, *both* the neural networks and the datasets are designed such that *we know which features are relevant* to the network output. Thus, we obtain the *ground truth attribution* in this synthetic environment. We leverage this information for the faithfulness test by measuring the alignment between the ground truth attribution and attribution maps (Section 4). If an attribution method fails to pass the faithfulness test in the simulated setup, its performance in more complex scenarios can also be suboptimal. A controlled environment can also be used to study the behavior of attribution methods under various circumstances by adjusting or ablating variables to simulate different scenarios. With the help of proposed synthetic environments, we examine a broad range of attribution methods and investigate the impact of several crucial factors, including the choice of baseline and superpixel segmentation (Section 5). We make several observations from the test results and provide suggestions for improving their attribution performance. Furthermore, we show how the controlled environment can be used to analyze perturbation-based faithfulness evaluation metrics (Section 6).

## 2 RELATED WORK

Since the advent of deep neural networks (Krizhevsky et al., 2012; Simonyan & Zisserman, 2015; He et al., 2016), understanding how these complex models make predictions came under the spotlight (Simonyan et al., 2013). One approach is a simplified view of identifying features relevant to a prediction, known as **feature attribution**. Initial efforts to perform feature attribution focused on linear models (Simonyan et al., 2013; Bach et al., 2015) and backpropagation (Springenberg et al., 2015; Zeiler & Fergus, 2014). Subsequently, more principled approaches emerged, such as backpropagating output differences with respect to a reference input (Bach et al., 2015; Shrikumar et al., 2017) and axiomatic methods (Sundararajan et al., 2017; Lundberg & Lee, 2017) inspired by the Shapley value (Shapley, 1953). Meanwhile, intuitive approaches probing internal states (Selvaraju et al., 2017; Schulz et al., 2020; Chattopadhay et al., 2018) and optimizing input masks were introduced (Fong et al., 2019; Zhang et al., 2021). With these advancements, **feature attribution evaluation** became a central question. Literature has proposed *sanity checks* testing whether attribution changes upon randomizing the network's weights (Adebayo et al., 2018) or if it is used on a different output class (Sixt et al., 2020). It also evaluates *faithfulness* by analyzing networks's output when perturbing input features based on their relevance (Samek et al., 2016; Ancona et al., 2018; Montavon et al., 2018; Hooker et al., 2019; Rong et al., 2022), or show theoretically if they are aligned with axioms outlining desirable properties (Sundararajan et al., 2017; Lundberg & Lee, 2017). Each perspective reveals different issues and properties (see Appendix B). A recent trend in evaluating *faithfulness* is designing datasets with known input-feature-output associations, enabling comparison between attribution and ground truth. Arras et al. (2022) generates datasets of shapes, Agarwal et al. (2023) generates synthetic graph datasets, and Zhou et al. (2022) proposes a dataset modification procedure to incorporate ground truth. However, these works do not guarantee that the network uses the intended ground truth associations. To address this problem, Khakzar et al. (2022) proposes a post-hoc solution of inverse feature generation by generating input features using the network. However, this direction faces challenges related to generated out-of-distribution input features. It is still unclear which parts of the generated features contribute and how they contribute.

## 3 A CONTROLLABLE ENVIRONMENT TO EVALUATE FAITHFULNESS

We aim to establish an environment where we know exactly which input features are relevant to the output of a model. This laboratory setup allows for testing the faithfulness of feature attribution methods. Within this setup, we can identify which attributions are not aligned with the true attribution of the network and understand the sources of failures. Prior to detailing the setup, we underline the necessity of designing both the *data* and the *network* to obtain the ground truth attribution.

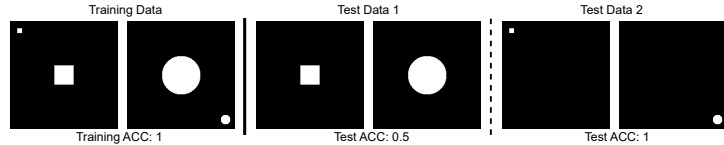

Figure 1: **Designing data is not enough.** Example on the neural networks not learning the designated ground truth features in the synthetic dataset. In this example, designed ground truth features are both objects in the center and on the edge. Even though the model can achieve $100\%$ accuracy, our test shows that the model only learns to use designed features at the corner and ignore the central ground truth features (more detail in Appendix A).

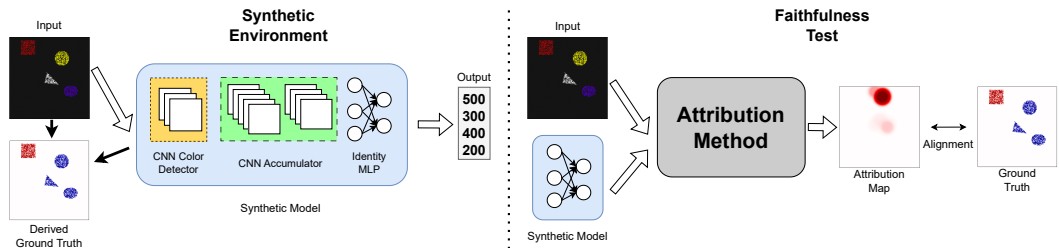

Figure 2: **Designing data and model** to set up a controllable environment for testing the faithfulness of attribution methods and analyzing their properties. To obtain the ground truth attribution, we explicitly design networks in tandem with inputs. The models follow conventional neural network designs and have sufficient complexity to evaluate various properties of attribution methods. The faithfulness test compares attribution results in the synthetic setting with the ground truth attribution.

To realize the controlled setup for evaluating explanations, one might design a dataset where the associations between input features and output labels are known: e.g., a simple dataset of squares and circles. Indeed, prior approaches (Arras et al., 2022; Zhou et al., 2022; Schuessler et al., 2021; Agarwal et al., 2023) design synthetic datasets and train a model on them. The underlying assumption is that a model learns the intended association in the dataset once the model achieves peak performance on the synthetic dataset during the learning process. However, it is unclear whether the trained network uses the entire square, straight lines, or just corners to identify the square. How can we know if a network uses an entire square for decision-making in this setup? This phenomenon is formulated as the *Rashomom effect*, which we rephrase in the following definition:

**Rashomon effect (Breiman, 2001)** *(multiplicity of equally good models) There exist many models under the same hypothesis class that can achieve equally good accuracy but use different information for inference.*

The Rashomon effect states that it is generally invalid to assume that a model with $100\%$ accuracy on a dataset $\mathcal{D}$ is ensured to learn the true labeling process. In reality, depending on the data distribution, we can have trained models that learn to ignore some features relevant to the labels. For instance, the model can learn other associations (e.g. use corners to classify a square). Specifically, neural networks tend to be over-parametrized for their learning tasks, hence making it harder to learn the true labeling function (Ba & Caruana, 2014; Frankle & Carbin, 2019). We provide empirical evidence for trained neural networks ignoring designed features (detailed setting in Appendix A). The result in Figure 1 shows a neural network, which achieves $100\%$ training accuracy but learns to solely use partial ground truth features designed for the dataset (objects at the edge). In a nutshell, a model can learn to perform correctly on the dataset but is not guaranteed to learn the intended ground truth features in the data. Furthermore, studies (Chen et al., 2019; Singh et al., 2020) demonstrate that attributions for a *learned* model are very fragile to spurious correlations (e.g. objects at the edge in above example). *Therefore, we manually design and set the weights of the neural network.*

By explicitly designing the data and the neural network according to specific properties, *we know the ground truth features (features used by the network)* in our synthetic setting. The knowledge of ground truth features can then be used for testing the faithfulness of feature attribution methods.

## 4    DESIGN OF DATA AND NEURAL NETWORK

We propose a modular setup where each component performs specific tasks and fulfills a purpose relevant to evaluating attribution methods. For the designs to facilitate the evaluation of feature attribution methods, we follow certain design principles. Firstly, the designs resemble real scenarios, such as image classification using a convolutional neural network (Figure 2). Designs that are similar to real-world cases can narrow the gap between real environments and synthetic environments. Hence, we can leverage synthetic environments to identify issues within attribution methods that are relevant to actual use cases. Furthermore, the design ensures that every ground-truth pixel is relevant and equally relevant. Specifically, with designed input data comprised of ground-truth (foreground) and baseline (background) pixels, designed neural networks have the following properties:

**Proposition 1** *(Sensitivity property) The addition/removal of **any** ground-truth pixel to/from the background affects the output of the designed neural network.*

**Proposition 2** *(Symmetry property) The addition/removal of any ground-truth pixel to/from the background **equally** affects the output of the designed neural network.*

The Sensitivity property implies that in this designed model and designed dataset setup, *every* ground truth pixel is relevant to the neural network output. The symmetry property implies that every ground truth pixel in the synthetic dataset should have *equal* relevance to the output of the synthetic neural network. In fact, these properties are aligned with the sensitivity and symmetry feature attribution axioms  (Sundararajan & Najmi, 2020; Sundararajan et al., 2017; Lundberg & Lee, 2017), which axiomatically define the relevance of a feature.

As our main environment, we design a setup that resembles real image classification tasks. The task is to identify the dominant color (in terms of number of pixels) within an input image (see Figure 2). The network first identifies predetermined colors and then counts the pixels for each. The designed dataset comprises color images, each containing $N_C$ patches of uniquely colored foreground objects and a distinct background color that is consistent across images, resulting in $N_C + 1$ colors in every image. The images are input to the model in RGB format. By treating each foreground color as a class, we formulate a multi-class task to predict the dominant color in an image. The designed classification model outputs softmax scores using logits for $N_C$ classes, where the logit of each class corresponds to the sum of pixels of the respective color. The softmax operation on the neural network output makes pixels of colors other than the target negatively contribute to the output of the target color (after softmax). Thus the ground truth will have both positively and negatively contributing features, enabling us to evaluate whether an attribution method is able to discern between positive and negative contributions. The design of the model and the dataset also follow the properties in the following propositions (Proposition  1 and  2). *Hence, we can infer that all pixels within the $N_C$ colored patches are relevant features.*

**Simulating "Unseen Data Effect"**    It is common that trained neural networks have unintended outputs given inputs that are not from the training distribution. However, since the data and every module (and weights) are set up manually, we know the expected behavior of the network for any input. Specifically, the network exclusively counts the number of pixels of predetermined colors, and other input values (colors) do not affect the network output. Through a neural operation (explained below), we can upgrade the setup to simulate the behavior of trained neural networks given data not previously seen by the model during training.

### 4.1    DESIGN DETAILS

The network is designed with an additional mode of behavior to imitate the behavior of trained neural networks. The default mode is that the model only responds to the predetermined colors and does not respond to other color values. The additional mode is that the model exhibits unpredictable behavior when given inputs that are not predetermined within the design (hence simulating the "Unseen Data Effect"). To realize these effects, the model has the following components. The first component is a CNN color detector responsible for detecting target colors and simulating Unseen Data Effects. Its output has dimensions of $(N_C + N_R) \times H \times W$, where $N_C$ denotes the number of target classes and $N_R$ denotes the number of *redundant* channels (the redundant channels are the neural

Figure 3: **Computational graph illustration** of our designed neural network modules. The left example shows a neural network of identifying number 5, and the middle example shows a simple color detector for detecting RGB value $(255, 127, 0)$. In these two cases, blue boxes symbolize neurons, with their respective computations indicated within the box. ReLU activation is applied after each neuron, which is omitted in the figure. The right example demonstrates CNN operations to achieve accumulation using non-uniform kernel weights. More details can be found in Appendix C.

implementation of "Unseen Data Effect" simulation). For the first $N_C$ channels, the $i^{th}$ output map is the activation of the $i^{th}$ target color. We briefly explain how to implement color detection in neural networks. Firstly, we design a neural structure that can identify a specific integer number. For integer input, this structure can be defined as

$$I_N(x) = \text{ReLU}(I_{>N-1}(x) - I_{>N}(x))$$
$$\text{where } I_{>i}(x) = \text{ReLU}(\text{ReLU}(x - i) - \text{ReLU}(x - i - 1)). \tag{1}$$

Given a color to be detected that is in RGB format $(R, G, B)$, where $R$, $G$, and $B$ are integers within $[0, 255]$. For a pixel with intensity $(r, g, b)$, the color detection mechanism shown in Figure 3 is:

$$C(r, g, b) = \text{ReLU}(I_R(r) + I_G(g) + I_B(b) - 2). \tag{2}$$

Here, the number identification functions $I_R$, $I_G$, and $I_B$ each detect a predefined component of an RGB value and are implemented as shown in Equation 1. Hence, we have $C(r, g, b) = 1$ for $r = R, g = G, b = B$, and $C(r, g, b) = 0$ otherwise. The remaining $N_R$ redundant channels activate on any other colors not among the $N_C + 1$ colors defined in our dataset. Specifically, if any pixel has a color that is neither a target color nor the background color, all $N_R$ redundant channels will activate at the position of this pixel. The activation mechanism of redundant channels is implemented as

$$R(r, g, b) = \text{ReLU}(-\sum C_i(r, g, b) + 1). \tag{3}$$

Consequently, $R(r, g, b) = 1$ if all $C_i(r, g, b) = 0$, and $R(r, g, b) = 0$ if $C_i(r, g, b) = 1$ for any $i$. Following the color detector, we have a CNN module that accumulates activation of the first $N_C$ channels respectively. We realize CNN accumulation in two settings, one with uniform weights in CNN, the other setting with non-uniform CNN weights. Figure 3 illustrates the working principle of pixel accumulation using CNN with non-uniform weights (more details in Appendix C.2). The remaining $N_R$ redundant channels have random connections to the input of this CNN module. Therefore, the CNN accumulation module will have unexpected outputs if the input images contain any color that is not seen in the training dataset. Lastly, we have an MLP that performs identity mapping, as the output of the CNN module already provides the logits of each target color for normal inputs. The rationale behind adding an identity MLP module is to preserve the conventional model architecture in image classification, where a model is typically designed as a CNN followed by an MLP head.

**Other synthetic settings** We provide other synthetic settings in Appendix D. The incentive for having additional synthetic settings is to cover more use cases, for instance, gray input images, single output models, and other model weight design schemes (e.g., modulo computation).

## 5 FAITHFULNESS TEST OF ATTRIBUTION METHODS IN ATTRIBUTIONLAB

In this section, we deploy the designed environment to test the faithfulness of attribution methods and analyze their various aspects. To provide an intuition about the behavior of various attribution methods, we visualize them in AttributionLab in Figure 4. This figure reveals a lack of consensus among attribution methods. To quantitatively test the alignment between attribution maps and ground truth masks, we use precision, recall, and $F_1$-score. However, since attribution maps usually have continuous pixel values, we adjust the metrics accordingly. Given a

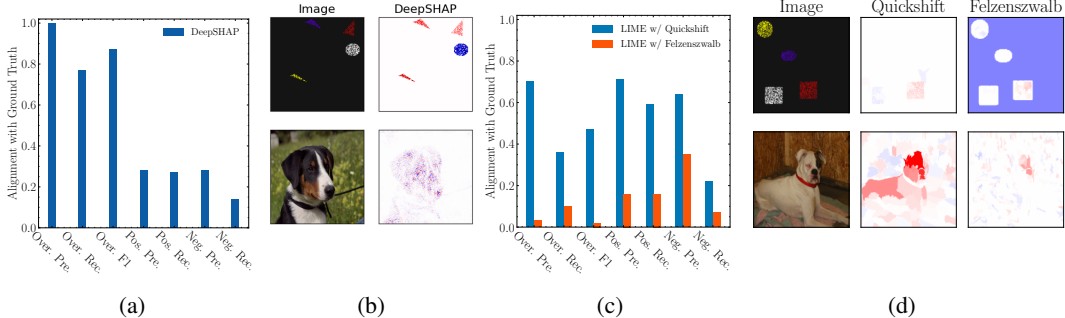

Figure 4: **Attributions in AttributionLab**. The leftmost image visualizes the sampled data. The second leftmost image shows the ground truth attribution, where red and blue denote positive and negative attribution to the target class, respectively. The sample is randomly selected. More samples can be found in Appendix F

Figure 5: (a) Faithfulness test of **DeepSHAP** in AttributionLab. (b) Visual example of DeepSHAP in synthetic and real-world environments. According to (a) and (b), DeepSHAP correctly highlights the foreground pixels. Nevertheless, it assigns both positive and negative attribution to these pixels, even when they have similar colors and close spatial locations. (c) Test result of **LIME** in AttributionLab. (d) Visual example of LIME in synthetic and real-world environments. Results in (c) and (d) show that the performance of LIME strongly depends on the segmentation result. [(*Pos.*) denotes the faithfulness test on positive attribution, while (*Neg.*) stands for the test on negative attribution, and *Over.* means the test on overall attribution neglecting the sign of attribution. *Pre.* and *Rec.* denote precision and recall, respectively.]

set of features $\mathcal{F}$ indexed by the index set $\mathcal{J} = \{j \in \mathbb{N}^+ | 1 \leq j \leq |\mathcal{F}|\}$, let $a_j$ be the attribution value of the $j^{th}$ feature, and $g_j$ be the ground truth value of the $j^{th}$ feature, the (soft) precision is calculated as $Pr(\mathcal{F}) = \sum_{j \in \mathcal{J}} |a_j \cdot g_j| / \sum_{j \in \mathcal{J}} |a_j|$, while the recall is defined as $Re(\mathcal{F}) = \sum_{j \in \mathcal{J}} |a_j \cdot g_j| / \sum_{j \in \mathcal{J}} |g_j|$. We normalize the attribution maps to $[-1, 1]$ to constrain the recall within the range of $[0, 1]$, as the ground truth masks are binary. Given the precision and recall, we can easily calculate the $F_1$-score as well. In multi-class classification, signed ground truth information is available, outlining the positive and negative contributions of features to the target class. To test attribution methods with signed ground truth, we separately compute the precision, recall, and $F_1$-score of the positive and negative portions of the attribution maps using their corresponding ground truth masks. Furthermore, we test the entire attribution map using an unsigned union of both positive and negative ground truth: the *overall ground truth*. This test takes into consideration all features that contribute to decision-making, ignoring the sign of attribution values.

Furthermore, we employ these attribution methods on an ImageNet (Deng et al., 2009)-pretrained VGG16 (Simonyan & Zisserman, 2015) and check if the test result can generalize to the real world. Specifically, we present visual samples from various attribution methods to affirm their consistent behaviors as observed in synthetic setups.

## 5.1 FAITHFULNESS TEST OF DEEPSHAP

DeepSHAP (Lundberg & Lee, 2017) is designed to identify both positively and negatively contributing features. Hence, we test its faithfulness by comparing the positive attribution with positive ground truth features (GT), negative attribution with negative GT, and overall attribution with overall GT by only considering the magnitude of attribution values. Figure 5a shows the test result of DeepSHAP in the synthetic environment. The overall precision, recall, and $F_1$-score reveal that DeepSHAP performs well in locating the contributing features when we disregard the sign of at-

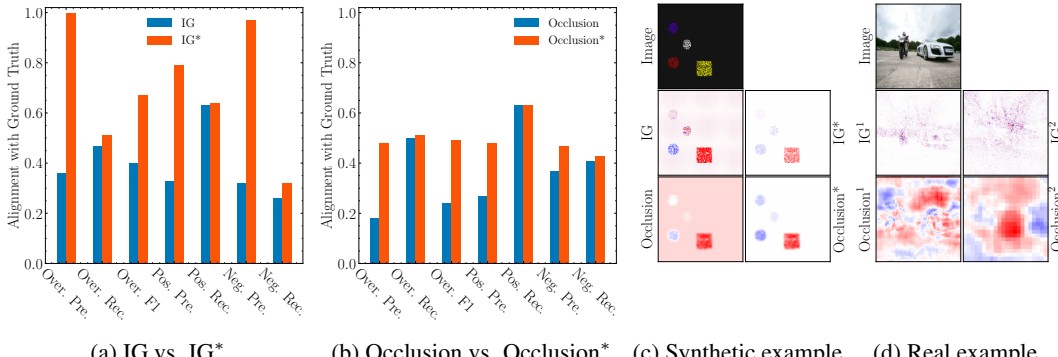

Figure 6: **IG and Occlusion with diverse baselines**. (a-b) IG* and Occlusion* employing the ground truth baseline display substantial enhancement in faithfulness test. (c) Comparative visualization of attribution maps, created with and without the utilization of the ground truth baseline. (d) Attribution maps generated on ImageNet. The superscripts signify the use of distinct baselines. Notably, the attributions highlight different foreground objects when employing different baselines.

tribution. However, the low precision and recall of positive and negative attribution suggest that DeepSHAP encounters difficulty in discerning whether a feature contributes positively or negatively to the target class. We further corroborated this issue by the visual results in Figure 5b, which show the phenomenon in both synthetic and real-world scenarios. In the ImageNet example shown in the image, we observe both positive and negative attribution on the *foreground* object, especially on some pixels that are closely located and have similar colors. However, most of these highlighted pixels are likely to be relevant to the target class of the image (though we can't ascertain which pixels the model is using). DeepSHAP is shown not to conform with Shapley value axioms (Sundararajan & Najmi, 2020). However, this claim does not tell us how faithful DeepSHAP is. Our results suggest that DeepSHAP can be used to identify all relevant features for the model without considering the sign of attribution values.

## 5.2 FAITHFULNESS TEST OF LIME

LIME (Ribeiro et al., 2016) requires the input image to be segmented into superpixels, it then treats all pixels within a superpixel as a single feature. Consequently, the resulting attribution map can be influenced by the segmentation step. To investigate the impact of segmentation, we utilize both Quickshift (Vedaldi & Soatto, 2008) and Felzenszwalb (Felzenszwalb & Huttenlocher, 2004) segmentation algorithms and test the faithfulness of the resulting attributions. Figure 5c reveals noticeable difference between the outcomes derived from these two segmentation techniques. Figure 5d provides additional visual evidence of these differences in real-world scenarios. In LIME, the attribution of a superpixel conveys the aggregate predictive influence of that segment. This "averaged" attribution is then broadcasted across all pixels within the superpixel. A more fine-grained attribution map requires a finer segmentation process. Hence, LIME requires prior knowledge regarding the ground truth features, that is, which pixels belong together and which ones are independent.

## 5.3 FAITHFULNESS TEST OF INTEGRATED GRADIENTS (IG), AND OCCLUSION

Previous research (Sturmfels et al., 2020) has revealed that IG (Sundararajan et al., 2017) is sensitive to the choice of baseline. While testing IG with a proper baseline remains a challenge in the real world, our controlled experimental setup provides the unique advantage of access to the true baseline. We introduce this baseline-accurate variant of IG as IG*, which uses the true baseline corresponding to background color $(20, 20, 20)$ in synthetic images. Despite that the visual difference between the true baseline and the widely used baseline $(0, 0, 0)$ is small, Figure 6a reveals a significant enhancement in the precision of IG*. We see that sensitivity to baseline is not an ignorable issue. Analogous to IG*, we introduce Occlusion* that employs the true baseline. Figure 6b also demonstrates a notably improved precision of Occlusion*. Figure 6c and Figure 6d further illustrate the sensitivity of these methods to the choice of baseline. Based on these empirical find-

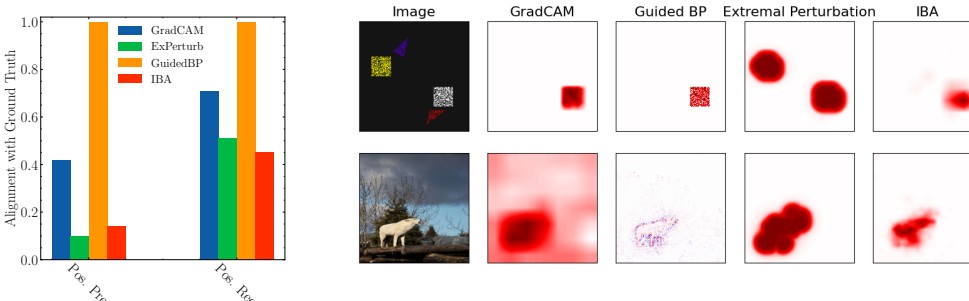

(a) Faithfulness test results           (b) Visual examples in synthetic setting and ImageNet

Figure 7: **Faithfulness test and visual examples** for GradCAM, GuidedBP, ExPerturb, and IBA. Some effectively identify positively contributing features, but none successfully discern negatively contributing features, inherent to their design limitations. GradCAM and IBA exhibit blurriness upon resizing, and the performance of ExPerturb is markedly influenced by the perturbation area.

ings, we underscore that one potential direction for enhancing these methods is the determination of an accurate baseline, even finding the baseline for each image separately (e.g., through generative models). Additionally, the performance of Occlusion* still remains less than optimal. One reason is that Occlusion's efficacy is influenced by the size of the sliding window used for pixel perturbation. Determining an appropriate perturbation area requires prior knowledge of both the dataset and the model structure, which is sometimes challenging to acquire in real-world applications.

## 5.4 FAITHFULNESS TEST OF GRADCAM, GUIDEDBP, EXPERTURB, IBA

GradCAM (Selvaraju et al., 2017) and GuidedBP (Springenberg et al., 2015) demonstrate high recalls on identifying *positively* contributing pixels. However, they do not identify pixels that negatively contribute to the target class, as evidenced by Figure 4 and Figure 7. This is because these methods typically initiate backpropagation from the logits before the softmax. The negatively contributing pixels do not influence backpropagation from the target class logit. Additionally, Grad-CAM resizes the attribution map to match the dimensions of the input. This resizing operation introduces blurriness (Figure 7b), which consequently diminishes its precision. As depicted in Figure 7a, ExPerturb (Fong et al., 2019) displays higher recall compared to its precision. This reduced precision can be attributed to the improper choice of the perturbation area constraint. In addition, Figure 4 and Figure 7b show that IBA (Schulz et al., 2020) can locate predictive features. However, the sigmoid used in the information bottleneck leads to only non-negative attribution.

## 6 FAITHFULNESS OF EVALUATION METRICS

There is another perspective through which faithfulness can be evaluated. We can perturb pixels in the image based on attribution values and observe how the output of the neural network changes. If the output change is aligned with the attribution values, then the attribution is considered faithful from this perspective. For instance, Insertion and Deletion (Samek et al., 2016) progressively insert or remove pixels, starting from the pixel with the highest attribution to the lowest attribution. In contrast to perturbing pixels according to a specific order, Sensitivity-N (Ancona et al., 2018) randomly selects $N$ pixels to perturb, after that measuring the correlation between the change in prediction and the sum of attribution of the perturbed pixels.

However, there is a fundamental issue lurking within this perspective on faithfulness. A significant output change can result from the network not having seen the new input resulting from perturbation rather than the perturbed pixel being important. In the controlled environment of AttributionLab, we can observe how sensitive these evaluation metrics are to Unseen Data Effect. That is when the pixels are perturbed during evaluation to another value for which the network is not programmed, a scenario that happens easily in practice. We conduct experiments using these metrics on two AttributionLab setups to assess how this phenomenon (Unseen Data Effect) affects their performance. Figure 8 shows that the order of feature attribution methods in terms of performance changes be-

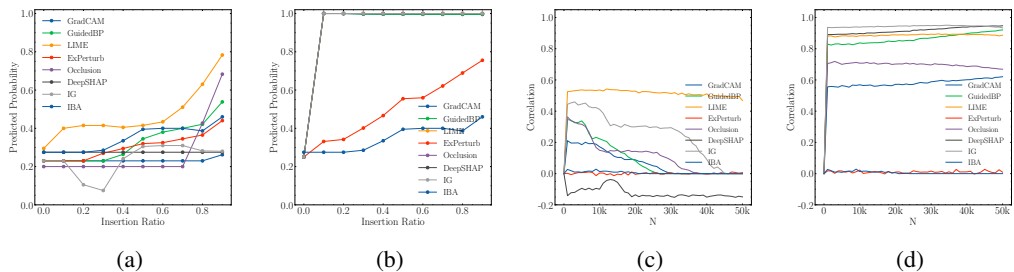

(a)  (b)  (c)  (d)

Figure 8: **Sensitivity of perturbation-based evaluation metrics to Unseen Data Effect**. We test these metrics on the two AttributionLab setups to observe the effect of this phenomenon on these metrics. (a) and (b) show the insertion metric results with and without Unseen Data Effect, respectively. (c) and (d) show the sensitivity-N metric results with and without the Unseen Data Effect. (in all metrics, a higher curve is better.) It is observed that with the presence of the Unseen Data Effect, the order of attribution methods changes in these metrics.

Table 1: **Spearman's rank correlation**. The rankings of attribution methods are measured on perturbation-based metrics and the designed ground truth, respectively. In the presence of the Unseen Data Effect, these metrics show significant deviation from ground-truth-based evaluations.

| Model has "Unseen Data Effect" | Insertion | Deletion | Sensitivity-N |
|:---:|:---:|:---:|:---:|
| Yes | 0.02 | 0.47 | 0.65 |
| No | 0.42 | 0.61 | 0.81 |

tween the two scenarios for both insertion and Sensitivity-N metrics. To confirm this observation, we compare the performance rankings of attribution methods in these metrics with performance rankings established from the ground truth. Therefore, we first rank the attribution methods using the $F_1$-score derived from *positive* ground truth attribution. Subsequently, we compute Spearman's rank correlation between the rankings, as demonstrated in Table 1. Additional experimental results are shown in Appendix F. The first row of Table 1 reveals that, in the presence of Unseen Data Effect, the metrics display substantially less consistency with the ground-truth-based evaluation. This inconsistency arises because Unseen Data Effect can result in unexpected predictions and inaccurate estimation of changes in model output. Hence, using model output scores as metrics may not accurately report the performance of attribution methods in the presence of Unseen Data Effect.

## 7    CONCLUSION AND FUTURE WORK

In this work, we propose a novel, controlled laboratory setup for testing feature attribution explanations. The crux of our approach is the implementation of a paired design, i.e., manually programming the neural network and designing the dataset. We test feature attribution explanations by simulating various conditions such as inappropriate baseline values for attribution methods and different segmentation masks as input of attribution methods, demonstrating their significant impact on attribution performance. We observe that the sensitivity to the baseline of Shapley value methods (SHAP and Integrated gradients) due to Unseen Data Effect is not an ignorable issue. We show that the Unseen Data Effect problem inherent in perturbation-based evaluations can have negative effects on evaluation findings. Our proposed synthetic environment can empower future research by identifying potential failure modes of attribution methods in a trustable, controlled environment. This could help researchers address these issues before deploying these methods in real-world settings, thus contributing to the development of more reliable and effective attribution methods. There are several potentials for future exploration. The community could develop additional settings to address a broader spectrum of potential issues in attribution methods. In addition, constructing more complex synthetic models and datasets would better approximate real-world scenarios. Furthermore, expanding this methodological approach to other data modalities and architectural designs could provide a more comprehensive understanding of feature attribution explanation methods.

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

# Appendix

## CONTENTS

# A EXPERIMENT DETAILS FOR FIGURE 1

The experiment result is shown in Figure 1. In this experiment, we design a synthetic dataset that comprises two classes. In each class, we only construct one image as the only data point. Each image has two visual components in the image. The class 0 image has one rectangular object in the center and another smaller rectangular object at the top left edge, while the class 1 image has one round object in the center and another round object at the lower right edge. We consider two components as the ground truth of the task. After training on the data, the model can achieve 100% accuracy. Then, we split every training image into two separate images, where one image contains the center object, and the other image contains the object at the edge. The split results are Test Data 1 and Test Data 2 in Figure 1. By using Test Data 1 and Test Data 2 as test images for the trained model, we observe that the model still achieves 100% accuracy for Test Data 2, but has 50% accuracy (which is random guess in 2-class case) for Test Data 1. Hence, we confirm that the trained model only learned to rely on edge objects for decision-making and completely ignored the centered object.

# B EXTENDED RELATED WORK

## B.1 FEATURE ATTRIBUTION

This section provides a taxonomy of several feature attribution methods.

### B.1.1 GRADIENT-BASED

The early methods relied on a linear approximation of the neural network around the input point. Simonyan et al. (2013) assumed the model to be linear (inspired by the piecewise linearity of ReLU) and proposed using the gradient of output with respect to input as a saliency map. This work coined the term "saliency map" for attribution, and many following works used the same nomenclature. The gradient of neural networks with respect to images looked "noisy". Therefore, there were attempts to remove the "noise" from the images. One of these methods averages the gradient over a neighborhood around the input to smooth out the gradient. The SmoothGrad Smilkov et al. (2017) method adds Gaussian noise to the image and generates multiple samples (in a Gaussian neighborhood of the input). Then computes the gradient and averages it. Other works proposed backpropagating only positive values to remove the "noise". Deconvolution Zeiler & Fergus (2014) and subsequently Guided Backpropagation Springenberg et al. (2015) and Excitation BackProp Zhang et al. (2018) work based on backpropagating positive values.

### B.1.2 LATENT FEATURES

The most famous work from this category is CAM/GradCAM Selvaraju et al. (2017); Zhou et al. (2016). The method does a weighted average of activation values of the final convolutional layer. The weights are selected based on their gradient values. Another work Khakzar et al. (2021) computes the input attribution based on the contribution of neurons to the output. This is achieved by computing the gradient of critical pathways. Another method restricts the flow of information in the latent space Schulz et al. (2020) and proposes information bottleneck attribution (IBA).

### B.1.3 BACKPROPAGATION OF RELEVANCE

The most important works within this category are LRP Bach et al. (2015), and DeepLift Shrikumar et al. (2017). The LRP method provides a general framework for backpropagating the relevance to input while satisfying a conservation property (similar to Kirchoff's law of electricity). The relevance entering a neuron equals the relevance coming out. There are several rules of backpropagation within the LRP framework, each with its own properties. DeepLift proposes a chain rule for propagating output differences to the input. An improved version of DeepLift is the DeepSHAP Lundberg & Lee (2017) method, which we use in the experiments.

### B.1.4 PERTURBATION-BASED METHODS

Another category of methods performs attribution by perturbing the input. The most straightforward way is to occlude/mask pixels (or patches of pixels) within the image and measure the output difference Zeiler & Fergus (2014); Ancona et al. (2018). The output difference reflects the importance of the removed feature. It is also possible to search for a mask on the input image. For instance, searching for the smallest region within the image which preserves the output Fong et al. (2019).

### B.1.5 SHAPLEY VALUE

The Shapley value Shapley (1953) itself is based on the occlusion of features. It is a notion from cooperative game theory for assigning the contribution of players to a game. We can consider the pixels within the image as players and the output of the network as the score of the game. The Shapley value is a unique solution that satisfies the symmetry, dummy, linearity, and completeness axioms altogether. It is of exponential complexity and, therefore, almost impossible to compute for large images and neural networks. Lundberg & Lee (2017) first proposed using the Shapley value (an approximation) for feature attribution in neural networks by introducing the SHAP and DeepSHAP (for neural networks) methods. Another method based on the Shapley value is the integrated gradients Sundararajan et al. (2017), which is the Aumann-Shapley value in the continuous domain Sundararajan & Najmi (2020).

### B.2 FEATURE ATTRIBUTION EVALUATION

There have been many efforts to evaluate feature attribution methods. Each of these methods evaluates the attribution through a different lens and reveals unique insights.

### B.2.1 ALIGNMENT WITH HUMAN INTUITION

Early feature attribution works Simonyan et al. (2013); Zeiler & Fergus (2014); Zhang et al. (2018); Bach et al. (2015) evaluated different saliency methods based on how they are aligned with what we humans think is salient. For instance, if a saliency method highlights a certain object, we might conclude that the network uses this object. However, the network may be using different features than ours. In this case, we wrongly consider the attribution method as correct (known as confirmation bias). Or in another case, the network might be using the object, but the saliency method might highlight something else. How would we know that the saliency is wrong? Several methods, such as the pointing game Zhang et al. (2018), made this visual evaluation systematic by comparing the saliency values with ground truth bounding box annotations. Even though human evaluation is flawed (e.g., due to confirmation bias), it is still useful for debugging and better understanding attributions.

### B.2.2 SANITY CHECKS

Another group of metrics evaluates the sanity of methods. They check if the methods possess specific properties that attribution methods must have. Adebayo et al. (2018) checks what happens to attributions when we replace the network weights with random values. It is surprisingly observed that some methods generate the same saliency values when the network is randomized. Another work Sixt et al. (2020) checks how the attribution method changes when it is applied to a different output. In the presence of multiple classes within the image, the attribution method is expected to point to the relevant features of the explained class.

### B.2.3 EVALUATION BY PERTURBATION

The intuition behind these methods is aligned with the perturbation-based feature attribution methods. If a feature is contributing to the output, its removal should affect the output. And the more the effect, the more important the feature. Based on this intuition, Samek et al. (2016) proposed perturbing/removing pixels based on their attribution score and plotting their effect as we keep removing them. However, one issue with this approach is that the output score might be due to out-of-distribution values and not the relevance of the feature. Another method Hooker et al. (2019) tries to remedy this issue by training the neural network from scratch on the perturbed dataset. In

other words, a certain percentage of each image within the dataset is perturbed based on attribution scores, and the network is trained on this new dataset. The method then computes the accuracies in different perturbation percentages.

### B.2.4 AXIOMATIC

Several axioms provide a framework to formalize the contribution of a feature Sundararajan & Najmi (2020); Lundberg & Lee (2017); Sundararajan et al. (2017). For instance, if the perturbation of a feature does not affect the output in any combination of features, then it is said to be a dummy feature. The dummy axiom demands that an attribution method should assign zero relevance to a dummy feature. Several other axioms reflect properties that we want the attributions to conform to. A combination of four properties, dummy, symmetry, linearity, and completeness, is only satisfied by the Shapley value Shapley (1953); Sundararajan et al. (2017). However, the Shapely value is not computable for large images and networks. Moreover, the proper reference value for removal must be chosen for computing the Shapley value. The complexity and the choice of reference values remain open problems. Moreover, for the existing methods, it is complex to theoretically show whether they conform to axioms as the methods are complex to analyze on neural networks. It is also shown that several axiomatic methods break the axioms in practical applications Sundararajan & Najmi (2020).

### B.2.5 ALIGNMENT WITH SYNTHETIC GROUND-TRUTH

An intriguing and promising solution is proposing synthetic settings to evaluate explanations. Several approaches propose generating datasets with known correlations between features and labels Arras et al. (2022); Agarwal et al. (2023); Zhou et al. (2022). However, there is no guarantee that the network will use the same features to predict the outputs. Therefore, we will not have a ground truth of which feature is relevant to the network. This category is the focus of this work. For further information, please refer to the main text.

## C IMPLEMENTATION DETAILS OF SYNTHETIC MODELS

In this section, we explain neural modules used in our synthetic model in finer detail.

### C.1 NEURAL NUMBER CHECKER

We show the design of a neural number checker, which is used in other synthetic neural networks. The number checker is designed to take integer numbers as input. Furthermore, this module should only activate on one predefined number by returning $1$, otherwise, it should always return $0$. To achieve number checking in neural networks, we first design a network that examines whether a number is greater than a predefined number. For this task, we design

$$I_{>i}(x) = \text{ReLU}(\text{ReLU}(x - i) - \text{ReLU}(x - i - 1)). \tag{4}$$

Figure 9 shows an example of this network. $I_{>i}(x)$ returns 1 if x is greater than $i$, and returns 0 otherwise.

With the help of $I_{>i}(x)$, we can further construct a neural network that performs number identification. For this purpose, we constrain the input to this network to be integers. Then, number identification can be achieved by performing

$$I_N(x) = \text{ReLU}(I_{>N-1}(x) - I_{>N}(x))). \tag{5}$$

The logic behind this design is that if $x$ is larger than both $N - 1$ and $N$, then the output is $0$. However, if $x$ is larger than $N - 1$, but not larger than N $N$, $I_N(x)$ returns 1. For natural number inputs, only $N$ suffice the condition for outputting 1. Figure 10 shows an example network $I_5(x)$.

### C.2 CNN ACCUMULATOR

In this section, we discuss how to construct CNNs that can perform pixel number accumulation. We design two types of CNNs for the accumulation task. One type of CNNs is initialized with uniform

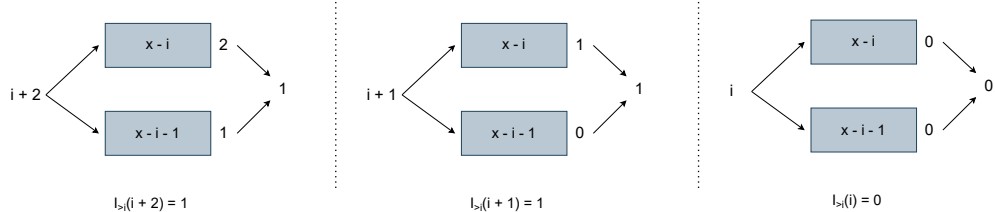

Figure 9: Example computational graph of $I_{>i}(x)$. Blue boxes symbolize neurons, with their respective computations indicated within the box. ReLU activation is applied after each neuron, which is omitted in the figure. We can set parameters depending on the $i$ value. We show three examples in the figure to show that this neural network structure activates by returning 1 when the input is greater than $i$, otherwise, it outputs 0.

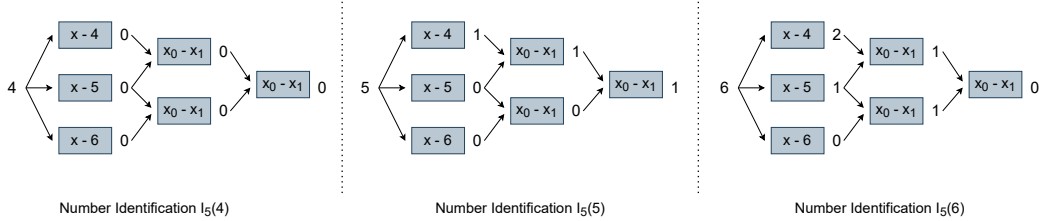

Figure 10: Example computational graph of $I_5(5)$. Blue boxes symbolize neurons, with their respective computations indicated within the box. ReLU activation is applied after each neuron, which is omitted in the figure. We show three cases where the input is 4, 5, and 6. For natural numbers as inputs, this network only returns 1 when the input is exactly 5. We can alter parameters to enable the identification of any natural numbers.

weights, while the other type can have non-uniform weights. We design these two types to show that our design can have both simple and complex CNN structures.

*Uniform weight initialization:* Realizing a CNN for pixel accumulation is intuitively simple. For a single-channel input, one can have a CNN module with multiple layers. Where each layer has only one kernel that sums up some neighboring pixels of the input. To realize this, the CNN kernel can have uniform weights of 1, and have a stride size equal to its kernel size. By applying multiple such CNN layers, we can sum up all pixels in the input. Figure 11 shows an example of a uniform CNN layer for accumulating pixels.

*Non-uniform weight initialization:* The above example shows the construction of a CNN accumulator using uniform weights. However, due to random initialization and randomness in training, it is unlikely that a trained model has uniform weights. Hence, we realize another version of CNN accumulator that uses non-uniform weights. For a non-uniform CNN accumulator, we cannot use a single CNN layer to perform the number accumulation task, as only a uniform CNN layer can carry this task. Instead, we develop a CNN accumulation block using two CNN layers. The first CNN layer in this block has $N$ kernels with kernel weight $w_{ij}$ for the $j^{th}$ index of the $i^{th}$ kernel. Thus, after applying this layer, we will have $N$ feature maps. The second CNN layer has a $1 \times 1$ CNN kernel, where its weight is defined as $m_i$ for the $i^{th}$ index. If the weight of these two layers satisfies $\forall j, \sum_i w_{ij} \cdot m_i = 1$, then the output still performs accumulation. Hence, we can initialize the accumulation block in a pseudo-random way, where we randomly set some of the weights, then calculate other weights that enable the CNN block to conduct accumulation. An example of this block is shown in Figure 12. In Figure 12, we simplify the design by using uniform weights for the second CNN layer. By subsequently connecting accumulation blocks, we can perform the accumulation task with a CNN module that has non-uniform kernel weights.

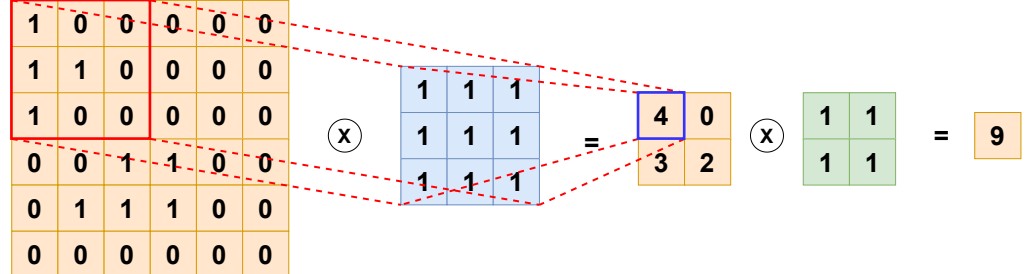

Figure 11: Example computation for an uniform CNN accumulator. ReLU activation is applied after each neuron, which is omitted in the figure. Given a single-channel input of the size $6 \times 6$, we can subsequently apply a uniform $3 \times 3$ CNN kernel and a $2 \times 2$ CNN kernel. The $1 \times 1$ output is the sum of input values. This example only has two CNN layers. For larger input sizes, we can design more CNN layers and kernels to perform accumulation.

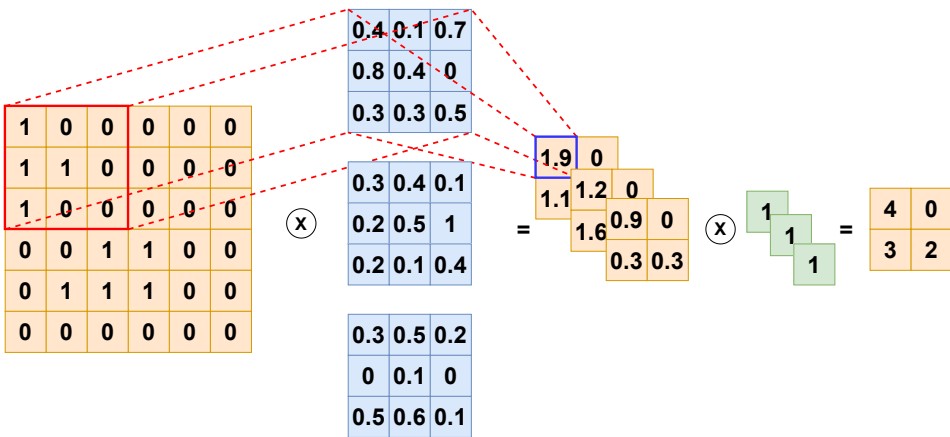

Figure 12: Example computation for a non-uniform CNN accumulation block. ReLU activation is applied after each neuron, which is omitted in the figure. The block consists of two CNN layers. In this example, the first layer has 3 $3 \times 3$ CNN kernels, the second layer has one $1 \times 1$ CNN kernel. One accumulation block can sum up neighboring pixels within a fixed range without having uniform weights.

### C.3 NEURAL MODULO MODULE

In this section, we provide an in-depth explanation of the designed model for modulo calculation. We explain the functionality of each MLP layer. The first layer of the module is

$$f_1(x) = \text{ReLU}(x, x - N, x - 2N, ..., x - \lceil \frac{U}{N} \rceil N), \tag{6}$$

where $U$ represents the maximal input value that the module can process, which can be scaled according to available computation resources. The incentive of applying $f_1(x)$ is to get $x \mod N$, $N + x \mod N$, $2N + x \mod N$, ..., until $x$. The calculation of $f_1(x)$ is valid for any natural number input that is not larger than $N$. The second layer is defined as

$$f_2(\vec{x}) = \text{ReLU}(x_0 - x_1, x_1 - x_2, ...). \tag{7}$$

Function $f_2(\vec{x})$ performs differentiation between subsequent outputs of $f_1(x)$. For all non-zero outputs of $f_1(x)$, their difference is always $N$. Furthermore, the smallest non-zero output is always $x \mod N$. Thus, we would have $f_2(f_1(x)) = (N, N, ..., x \mod N, 0, ..., 0)$ after propagating through the first two layers. At this stage, our output only comprises three possible numbers: $N$, $x \mod N$, and 0. Next, we eliminate the number $N$ in the output vector. This is accomplished by first checking if an output is $N$, then eliminating $N$, which is done by applying

$$f_3(\vec{x}) = \text{ReLU}(x_0, I_N(x_0), x_1, I_N(x_1), ...),\tag{8}$$

and

$$f_4(\vec{x}) = \text{ReLU}(x_0 - Nx_1, x_2 - Nx_3, ...).\tag{9}$$

$f_3(\vec{x})$ is responsible for checking all outputs after $f_2(\vec{x})$, if an output is $N$, it appends a flag number 1 after this output. Otherwise, if an output is not $N$, which is either $x \mod N$ or 0, it appends a flag number 0. $f_4(\vec{x})$ Subtracts each output of $f_2(\vec{x})$ with $N$ times its flag number obtained from $f_3(\vec{x})$. Hence, we have $f_4(f_3(f_2(f_1(x)))) = (0, 0, ..., x \mod N, 0, ...)$. Lastly, we have one layer

$$f_5(\vec{x}) = \text{ReLU}(\sum_{i=0} x_i),\tag{10}$$

to accumulate the output vector. Therefore, as shown in Figure 13, we can obtain the modulo calculation with

$$f_{\text{modulo},N}(x) = f_5(f_4(f_3(f_2(f_1(x))))) = x \mod N.\tag{11}$$

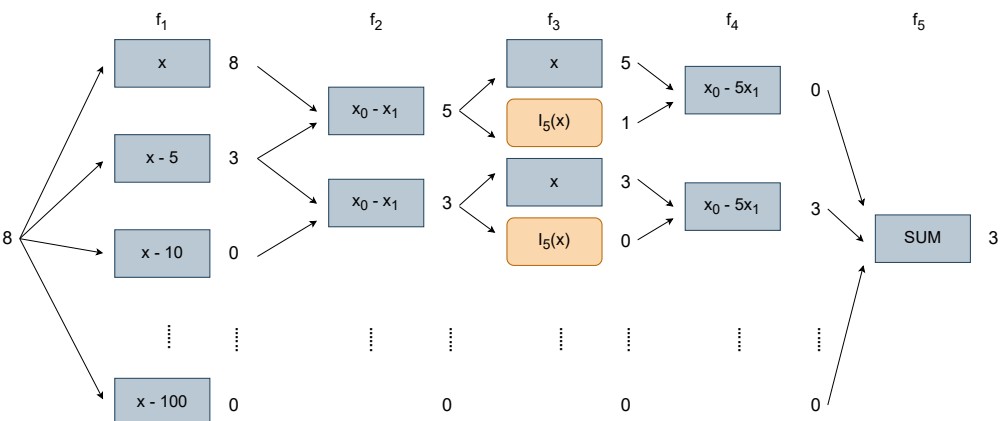

Figure 13: Example computational graph of $f_{\text{modulo},5}(8)$. Blue boxes symbolize neurons, with their respective computations indicated within the box. ReLU activation is applied after each neuron, which is omitted in the figure.

## C.4 NEURAL COLOR DETECTOR MODULE

This section describes how we design a CNN part that can detect certain colors. Since we want to check every pixel in an image, we design CNN modules with $1 \times 1$ kernels for color detection. For this module, we assume the input image is in RGB format, and the output of this module has dimensions of $N \times H \times W$, where $N$ denotes the number of colors we want to detect, $H$ and $W$ are the height and the width of the input image. Each of the $N$ output channels is responsible for the detection of a predefined target color. Specifically, if a pixel in the input corresponds to the $i^{th}$ target color the model aims to detect, then the neuron at the same position in the $i^{th}$ channel of the output should be 1. Otherwise, the corresponding output neuron should be 0. To accomplish this, the module should be able to check all three input channels and compare if the RGB values match the color to be detected. As discussed in Section 4, we leverage

$$C(r, g, b) = \text{ReLU}(I_R(r) + I_G(g) + I_B(b) - 2)\tag{12}$$

for detecting RGB values of pixels. $C(r, g, b)$ only outputs 1 if all three input values match the RGB value of the color to be detected. Illustration of $C(r, g, b)$ is shown in Figure 3. To enable

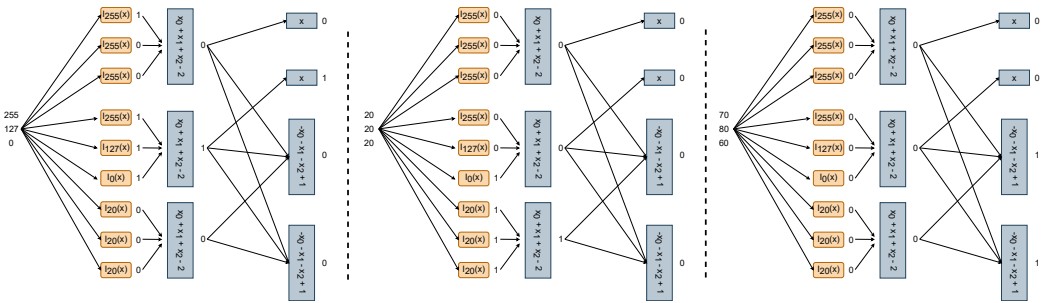

Figure 14: Example computational graph for color detection. Blue boxes symbolize neurons, with their respective computations indicated within the box. ReLU activation is applied after each neuron, which is omitted in the figure. This example shows a color detector with two redundant channels. The color detector is responsible for detecting target colors $(255, 255, 255)$ and $(255, 127, 0)$. Moreover, the color detector is programmed to ignore the background color, which is $(20, 20, 20)$. In the left case, one of the target colors is detected. Hence, the detector activates the target color channel. In the middle case, none of the target colors is detected. However, the input is the same as the predefined background color $(20, 20, 20)$. Subsequently, the input is considered in distribution, and no output channel is activated. In the right case, the input is not one of the target colors or the background color, so all redundant channels are activated in response to OOD inputs. We show the computation in fully connected NN layers. Nevertheless, color detectors are realized as CNN layers with $1 \times 1$ kernels in our implementation.

simultaneous detection of multiple target colors, we concatenate several $C(r, g, b)$ with their weights set for different RGB values. This is shown as the first two layers of the network in Figure 14.

We further elaborate on how to create redundant channels in our color detector design. Instead of outputting the activation map of the dimension $N \times H \times W$, the output can have the shape of $(N + R) \times H \times W$, where $R$ denotes the number of redundant channels. Redundant channels serve as OOD channels that only activates if OOD colors are present in an image. We define OOD colors to be any color other than our target colors and the predefined background color. To activate redundant channels, we use

$$R(r, g, b) = \text{ReLU}(-\sum C_i(r, g, b) + 1), \tag{13}$$

where $C_i(r, g, b)$ indicates color detection for different colors. If none of any $C_i(r, g, b)$ is activated, then $R(r, g, b)$ returns 1. In the example shown in Figure 14, the last two outputs are redundant channels.

## D   IMPLEMENTATION OF SINGLE-COLOR-MODULO SETTING

Given monochromatic images as input, we manually program a synthetic neural network to count the number of white pixels in the image (the Accumulator) and output the modulo $N$ of the total pixel count (the Modulo), where the divisor $N$ is a predefined number. The design of these images consists of multiple pixel patches of arbitrary shape within a constant background (reference value). The Accumulator and Modulo programs have *properties* tailored toward evaluating feature attributions: In our model design, the addition/removal of any ground truth (white) pixel to/from the background (black) equally affects the count of white pixels. Hence, we know each and every white pixel is equally relevant.

To develop a neural network model that can perform this task, we design a model that consists of two components. The first component is a Convolutional Neural Network (CNN) tasked with counting the total number of white pixels in the input image. This CNN accumulation module can be initialized with either uniform or non-uniform weights. To initialize this CNN with uniform weights, we can simply set all weights of CNN kernels to 1. We further elaborate on the initialization of non-uniform weights in Appendix C.2. The second part of the model is a multilayer perceptron (MLP) designed to perform the modulo operation. We formally define the modulo module using functions. Firstly, we leverage the number identification module $I_N(x)$ to further build up the neural modulo

module. The first layer of the module is $f_1(x) = \text{ReLU}(x, x - N, x - 2N, ..., x - \lceil \frac{U}{N} \rceil N)$, where $U$ represents the maximal input value that the module can process, which can be scaled according to available computation resources. Note that $f_1$ produces a vector output. The second layer is defined as $f_2(\vec{x}) = \text{ReLU}(x_0 - x_1, x_1 - x_2, ...)$. Thus, we would have $f_2(f_1(x)) = (N, N, ..., x \mod N, 0, ..., 0)$ after propagating through the first two layers. At this stage, our output only comprises three possible numbers: $N$, $x \mod N$, and 0. Next, we eliminate the number $N$ in the output vector. This is accomplished by applying $f_3(\vec{x}) = \text{ReLU}(x_0, I_N(x_0), x_1, I_N(x_1), ...)$, and $f_4(\vec{x}) = \text{ReLU}(x_0 - Nx_1, x_2 - Nx_3, ...)$. Hence, we have $f_4(f_3(f_2(f_1(x)))) = (0, 0, ..., x \mod N, 0, ...)$. Lastly, we have one layer $f_5(\vec{x}) = \text{ReLU}(\sum_{i=0} x_i)$ to accumulate the output vector. Therefore, as shown in Figure 3, we can obtain the modulo calculation with

$$f_{\text{modulo}, N}(x) = f_5(f_4(f_3(f_2(f_1(x))))) = x \mod N. \tag{14}$$

The complexity of the MLP model ensures that the synthetic model poses a moderate challenge for attribution methods. Further details regarding the dataset generation and model initialization can be found in Appendix C.1, Appendix C.3.

## E    EXPERIMENTS IN THE SINGLE-COLOR-MODULO SETTING

### E.1    SYNTHETIC DATASET GENERATION

**Image generation:** We place four non-overlapping patches at four randomly selected centers within each image, with each patch surrounded by Bézier curves. Pixel intensities in the patches are randomly sampled from the Bernoulli distribution $Bernoulli(0.5)$. To save the generated images in single-channel PNG format, we scale the pixel value of 1 to 255. Samples of generated images are shown in the first column of Figure 15.

**Ground truth label:** We predefine a modulus $N$ (default value 30). Let $s$ represent the sum of all 1s in a synthetic image; the ground truth label for the image is then determined as $s \mod N$. For example, if an image contains 100 pixels of 1 and $N = 30$, the ground truth label is 10.

**Ground truth features:** For white pixels in the image, pixels have a uniform ground truth feature importance of 1, whereas for black pixels, they have a uniform ground truth feature importance of 0.

We visualize some randomly selected samples in Figure 15, including the generated images, ground truth features, and attribution maps.

### E.2    HYPER-PARAMETERS OF ATTRIBUTION METHODS

In this subsection, we present the key hyperparameters for various attribution methods to ensure reproducibility in future studies. The visual results of these attribution methods are presented in Figure 15.

- GradCAM: GradCAM is attached to the sixth layer of the CNN accumulator, specifically at `accumulator.layers.5` in our implementation.
- GuidedBP: Backpropagation is performed from the scalar modulo output.
- LRP: Similar to GuidedBP, backpropagation is performed from the scalar modulo output.
- ExPerturb: A perturbation area of 0.1 is used, with Gaussian blurring applied for image perturbation. The Gaussian blurring sigma is set to 21.0.
- Occlusion: A sliding window of shape $(1, 5, 5)$ and strides $(1, 3, 3)$ is employed, along with an all-zero baseline.
- DeepSHAP: An all-zero baseline is used.
- IG: An all-zero baseline is applied.
- IG*: This method is equivalent to IG, as the ground truth baseline is 0.
- IBA: The information bottleneck is attached to the twelfth layer of the CNN accumulator, specifically at `accumulator.layers.11` in our implementation. The weight of the information loss is set to 20.

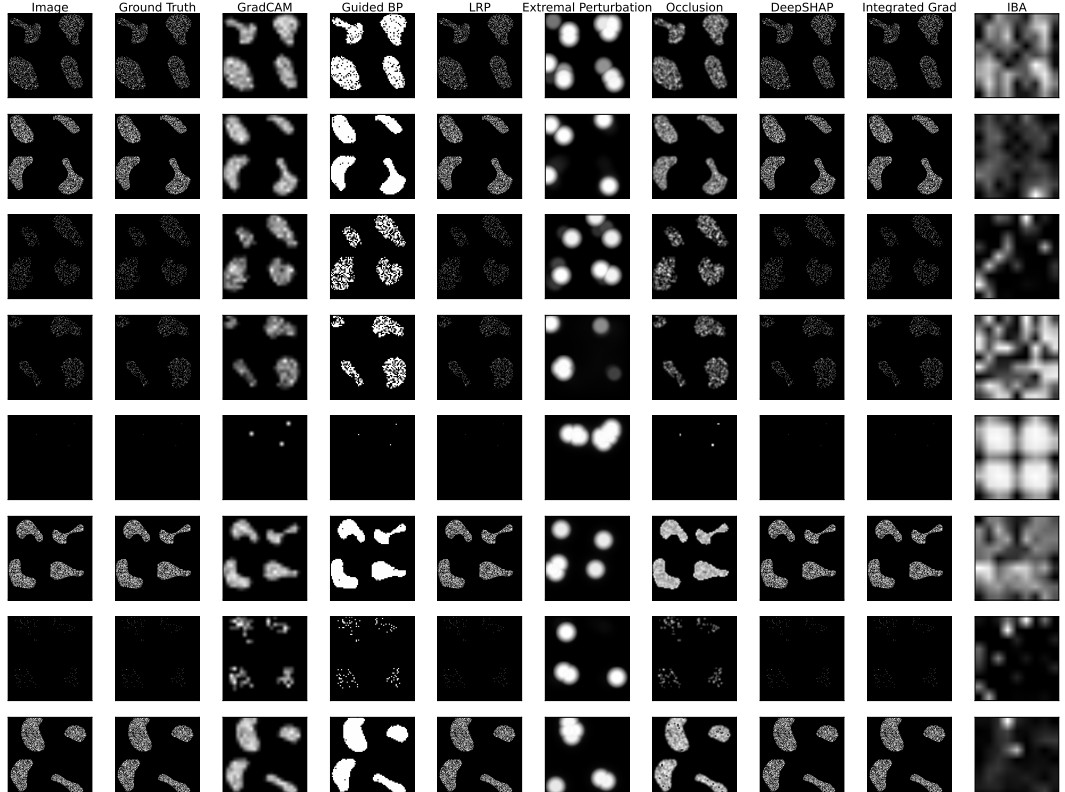

Figure 15: Visualization of images and attribution maps in single-color-modulo setting.

- Random: Random attribution values are sampled from a uniform distribution $\mathcal{U}(0,1)$.
- Constant: The attribution values are set to a constant value of $1.0$.

### E.3 GROUND-TRUTH-BASED EVALUATION

#### E.3.1 EXPERIMENTAL SETUP

We compute the *overall* precision, recall based on the synthetic ground truth (feature importance) masks. Note that some attribution method can return attribution values substantially greater than $1.0$, which are not in the same scale as the ground truth value $1.0$. In this case, if we apply the formula of recall in Section 3, we would get a recall greater than $1.0$. To avoid this undesirable outcome, we normalize the attribution map as follows:

$$t(a_j) = \begin{cases} \frac{a_j}{a_+} & \text{if } a_j \geq 0; \\ \frac{a_j}{|a_-|} & \text{else,} \end{cases} \tag{15}$$

where $a_j$ is the attribution associated with $j^{th}$ feature, $a_+ = \max_j a_j$ for all $a_j \geq 0$, and $a_- = \min_j a_j$ for all $a_j < 0$, respectively. The attribution value after normalization is within the interval $[-1, 1]$.

#### E.3.2 EXPERIMENTAL RESULTS

Figure 16 presents violin plots of overall precision and recall. The three horizontal bars within each plot correspond to maximum, average, and minimum values, respectively. These plots indicate that LRP, DeepSHAP, and IG (equivalent to IG$^*$ in this setting) achieve optimal precision and recall for nearly all data samples. Although GuidedBP obtains high recall for most samples, its average precision is suboptimal. This observation is supported by the visual results, where GuidedBP mistakenly identifies black pixels within the Bézier patches as contributing pixels. Furthermore, the low

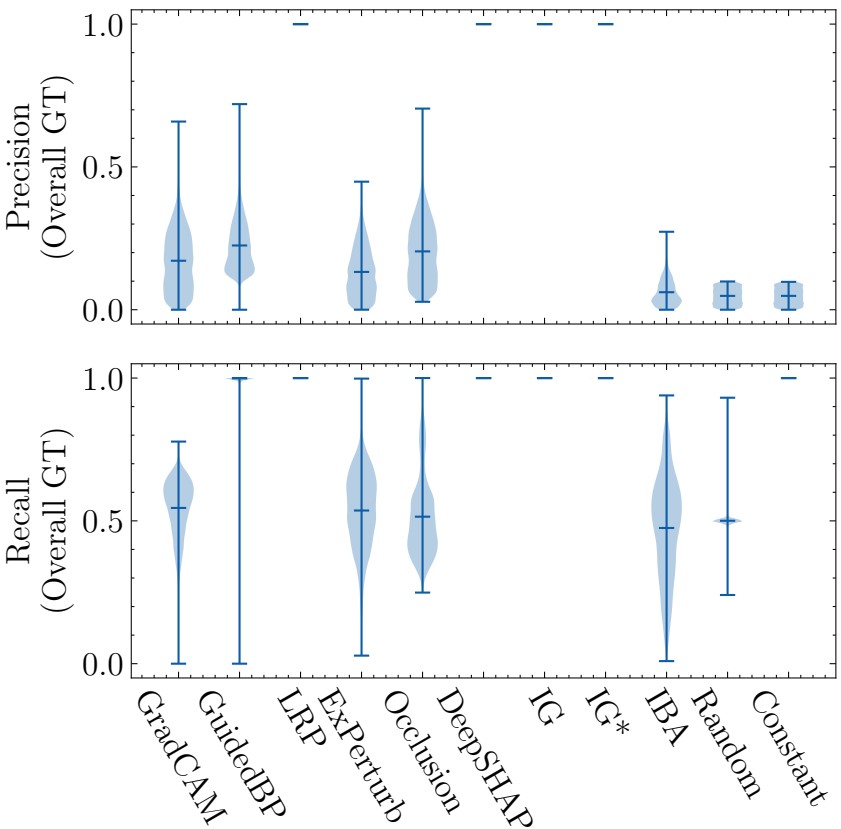

Figure 16: Ground-truth-based evaluation in the single-color-modulo setting. The maximal, minimal, and average values are marked with horizontal bars.

Table 2: Rankings of attribution methods in the single-color-modulo setting when evaluated with ground truth masks, Insertion/Deletion, and Sensitivity-N. Correlation denotes the Spearman's rank correlation with the $F_1$-score (computed using the overall ground truth masks).

| Attribution methods | Ranking | | | |
| --- | --- | --- | --- | --- |
| | $F_1$-score | Insertion | Deletion | Sensitivity-N |
| GradCAM | 6 | 6 | 6 | 6 |
| GuidedBP | 4 | 5 | 5 | 4 |
| LRP | 0 | 3 | 0 | 0 |
| ExPerturb | 7 | 7 | 7 | 7 |
| Occlusion | 5 | 4 | 4 | 5 |
| DeepSHAP | 1 | 0 | 1 | 1 |
| IG | 2 | 1 | 2 | 2 |
| IG* | 3 | 2 | 3 | 3 |
| IBA | 8 | 8 | 8 | 8 |
| Random | 10 | 10 | 9 | 10 |
| Constant | 9 | 9 | 10 | 9 |
| Correlation | — | 0.94 | 0.98 | 1.00 |

precision of methods such as GradCAM and IBA is often attributed to their process of performing attribution on a neural network's hidden layer and subsequently resizing and interpolating the attribution maps to match the input image's spatial dimensions. This additional post-processing step frequently leads to over-blurring of attribution maps.

Table 3: Insertion/Deletion evaluation results in the single-color-modulo setting. A higher Insertion AUC and lower Deletion AUC indicate a better attribution method.

| Attribution Method | Insertion AUC ↑ | Deletion AUC ↓ |
|---|---|---|
| GradCAM | 0.8827 | 0.1174 |
| GuidedBP | 0.9015 | 0.0983 |
| LRP | 0.9754 | 0.0246 |
| ExPerturb | 0.7845 | 0.2157 |
| Occlusion | 0.9151 | 0.0849 |
| DeepSHAP | 0.9754 | 0.0246 |
| IG | 0.9754 | 0.0246 |
| IG* | 0.9754 | 0.0246 |
| IBA | 0.6458 | 0.3526 |
| Random | 0.4996 | 0.5009 |
| Constant | 0.4447 | 0.4439 |

## E.4 INSERTION/DELETION

### E.4.1 EXPERIMENTAL SETUP

In the single-color-modulo setting, directly comparing modulo results between perturbed and original images is insufficient for determining true pixel contribution, as perturbing a specific number of pixels can produce the same modulo result. For instance, if the sum of 1s in perturbed pixels is divisible by the modulus $N$, the change in the modulo result is 0, suggesting no contribution from the perturbed pixels. To address this, we adapt the implementations of these metrics without compromising their correctness. We perturb pixels individually and compare model outputs at consecutive steps. Additionally, we use zero-intensity pixels as replacement pixels during the progressive perturbation process. Different modulo results indicate genuine pixel contribution and the attribution method's accurate identification of the contributing pixel at that step. Otherwise, the step is deemed a failure for the attribution method. We accumulate correct steps as a substitute for prediction variation in the original Insertion/Deletion or Sensitivity-N. In fact, the accumulated correct steps equal the number of perturbed pixels with value 1. To compute the AUC in Insertion/Deletion, we normalize the number of correct steps by dividing it by the total contributing pixels in the ground truth mask, yielding a range of $[0, 1]$.

### E.4.2 EXPERIMENTAL RESULTS

As illustrated in Table 3, the best performing methods are IG (or IG*), DeepSHAP, and LRP, achieving optimal Insertion AUC and Deletion AUC. As indicated by the Spearman's rank correlations in Tables 1 and 2, the Insertion/Deletion evaluation is highly consistent with the ground-truth-based evaluation. It is important to note that the optimal Insertion AUC in Table 3 is not 1.0, and the optimal Deletion AUC is not 0.0. This occurs because we remove one pixel at each step, and the best Insertion curve is obtained when all 1s in the image receive higher attribution than all 0s. In this case, the Insertion curve is a monotone increasing straight line followed by a saturated flat line at 1.0. Similarly, the optimal Deletion curve occurs when all the 1s are removed before all the 0s, resulting in a monotone decreasing straight line followed by a stagnant flat line at 0.0. Consequently, the AUC of the optimal Insertion curve is smaller than 1.0, and the AUC of the optimal Deletion curve is greater than 0.0. The optimal AUCs depend on the number of pixels with value 1 and the total number of pixels.

## E.5 SENSITIVITY-N

### E.5.1 EXPERIMENTAL SETUP

In standard Sensitivity-N, we randomly select $N$ pixels at each step. However, this approach is not applicable in the single-color-setting due to the same reason we explained previously in Insertion/Deletion. Therefore, following our dapted Insertion/Deletion in the single-color-modulo setting, we measure the model output change in an accumulative manner. Specifically, we need to compare

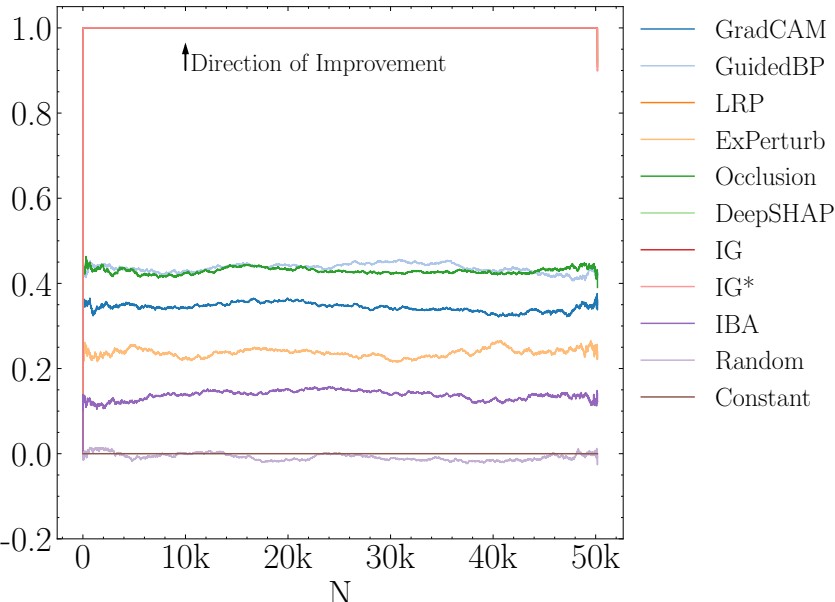

Figure 17: Sensitivity-N evaluation results in the single-color-modulo setting. $y$-axis represents the Pearson Correlation Coefficient between the model output change and the total attribution of perturbed pixels.

the modulo results of two consecutive steps. Note that we can determine whether the $i^{th}$ pixel is contributing only when it is the sole pixel in which the perturbed pixels at two consecutive steps differ. In other words, we need to randomly select an *additional* pixel to perturb compared to the previously perturbed pixels. Then, we record the model output change and the total attribution of perturbed pixels, respectively. We repeat the random selection for 100 times and concatenate the recorded output change and total attribution into two vectors, respectively. Next, the Pearson Correlation Coefficient is computed between these two vectors. Subsequently, we increase the number of perturbed pixels $N$ from 1 to the image size $224 \times 224$ and repeat the above procedure.

### E.5.2 EXPERIMENTAL RESULTS

For readability, we reproduce the Sensitivity-N figure from Section 6 in Figure 17. DeepSHAP, IG (or IG$^*$), and LRP achieve (near) optimal performance. This is consistent with the Insertion/Deletion results in Table 3 and ground-truth-based evaluation results in Figure 16.

## F EXPERIMENTS IN THE MULTI-COLOR-SUM SETTING

In this section, we present more experimental results and explanations as a supplement to Section 5 and Section 6. Specifically, we present additional experimental results on the Multi-color-sum setting.

### F.1 SYNTHETIC DATASET GENERATION

**Image generation:** This setting is designed to simulate a multi-class classification task. We randomly place four non-overlapping patches on an empty background image with a uniform intensity of $(20, 20, 20)$ (unsigned 8-bit `int`) in RGB format. Patch shapes are sampled from three categories: triangle, square, or circle. Additionally, patch sizes are randomly sampled. For each pixel within each patch, we independently and randomly sample a random variable from the $Bernoulli(0.5)$ distribution. If the sampled value is 1, the pixel is set to the background color; otherwise, it is set to a unique color associated with the corresponding patch. Ultimately, an image will contain four distinct foreground colors. Some randomly selected samples are visualized in Figure 18.

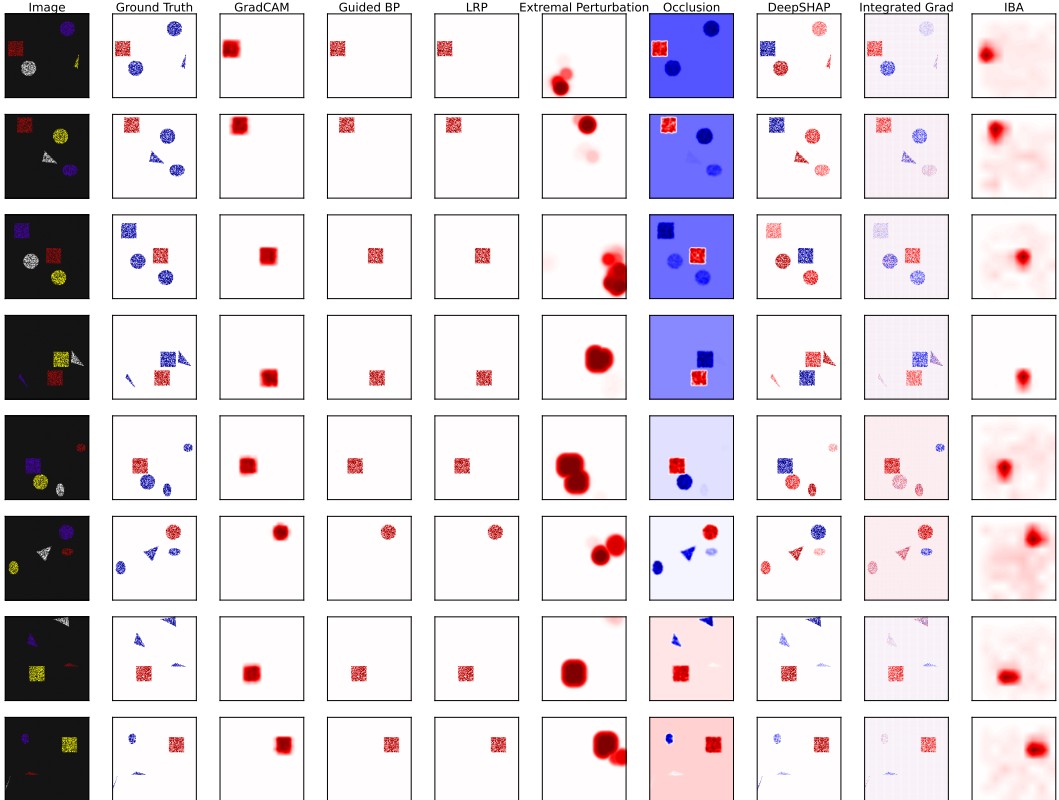

Figure 18: Visualization of images and attribution maps in multi-color-sum setting.

**Ground truth label:** The task is a multi-class classification task, where each class represents a non-background color. For each randomly generated image, the winning color in the image is the one with the most pixels of that color, and the image is labeled by the class of the winning color.

**Ground truth features:** The greater the number of pixels a color has, the more likely it is to become the winning color. Moreover, each pixel contributes equally when being counted. The corresponding output of a color increases by 1 if we insert a pixel of that color into the image. For a non-winning color, increasing its number of pixels raises the likelihood of it becoming the winning color, thereby decreasing the probability of the current winning color. Background pixels do not contribute to the current winning color, as they are not detected by the color detector. Based on these principles, we establish the ground truth feature contribution masks as follows: (1) all pixels belonging to the winning color have a uniform contribution of $1$; (2) all pixels belonging to the non-winning colors have a uniform contribution of $-1$; and (3) all background pixels have a uniform contribution of $0$. As shown in Figure 4, positively contributing pixels are displayed in red, negatively contributing pixels are displayed in blue, and non-contributing pixels are displayed in white.

### F.2 HYPER-PARAMETERS OF ATTRIBUTION METHODS

In this subsection, we present the key hyper-parameters of attribution methods employed in this setting:

- GradCAM: GradCAM is attached to the sixth layer of the CNN accumulator (specifically, `accumulator.layers.5` in the implementation).
- GuidedBP: Backpropagation is computed from the output after the softmax layer.
- LIME: The default segmentation method is Quickshift Vedaldi & Soatto (2008).
- ExPerturb: We select a perturbation area of $0.1$ and employ Gaussian blurring for perturbation.

- Occlusion: A sliding window of shape $(3, 5, 5)$ and strides $(3, 3, 3)$ is used, along with an all-zero baseline. Although we know the ground truth baseline, we do not use it here because the true baseline is difficult to access in real-world applications, and we aim to study the influence of an inappropriate baseline in the multi-color setting. Furthermore, the output change is measured at the layer before the softmax, as suggested by the original paper Zeiler & Fergus (2014).

- DeepSHAP: The output change is measured after the softmax layer, and an all-zero baseline is employed.

- IG: The output change is measured after the softmax layer, and an all-zero baseline is employed.

- IBA: The information bottleneck is inserted into the eighth layer of the CNN accumulator (specifically, `accumulator.layers.7` in the implementation). Additionally, the weight of information loss is set to 20.

### F.3 GROUND-TRUTH-BASED EVALUATION

#### F.3.1 EXPERIMENTAL SETUP

We normalize the attribution maps in the same way as in Equation 15. Then, the precision, recall, and $F_1$-score are computed based on three sets of the designed ground truth masks, denoted with overall, positive, and negative ground truth (GT), respectively. Note that not all attribution methods can return negative attribution.

#### F.3.2 EXPERIMENTAL RESULTS

Figure 19 displays the violin plots of precisions and recalls computed with different sets of GT. Several methods, such as GradCAM, GuidedBP, LRP, and IBA, only identify the pixels contributing to the label class, ignoring the negatively contributing pixels, as shown in Figure 18. Therefore, the recalls computed with positive GT for these methods are higher than the recalls computed with overall GT, as the overall GT includes both positively and negatively contributing pixels. Additionally, if we only consider the pixels of the label class, IBA exhibits superior performance by comparing the precision associated with positive GT in the multi-color-sum setting with the precision computed with overall GT in the single-color-modulo setting. A possible reason could be that IBA is formulated using variational methods, where the objective function is relaxed using cross-entropy. Cross-entropy is more suitable for multi-class classification in the multi-color-sum setting. To enhance performance in the single-color-modulo setting, an alternative version of the optimization objective for IBA might be required. Moreover, DeepSHAP, IG, and Occlusion necessitate knowing the ground truth baseline of the model, which is very difficult to acquire in real-world scenarios. In this study, we select a baseline $(0, 0, 0)$ that causes unpredicted model behavior, even though it is visually very close to the true baseline color $(20, 20, 20)$. In Figure 18, we observe that Occlusion is prone to breaking down, and IG assigns random positive or negative attribution to the background. If we select the correct baseline for IG, we can significantly enhance its precision, as demonstrated by IG$^*$ in Figure 19. Furthermore, DeepSHAP achieves high precision and recall when computed with overall GT. However, it often misidentifies positively contributing pixels as negative, and vice versa. Consequently, DeepSHAP exhibits low precision and recall when computed with positive or negative GT. This suggests that DeepSHAP can accurately locate the ground truth features but often yields the wrong sign of attribution.

### F.4 INSERTION/DELETION

#### F.4.1 EXPERIMENTAL SETUP

The implementation of Insertion/Deletion in the multi-color-sum setting is consistent with the original paper. Specifically, we progressively perturb the pixels with replacement pixels with intensity $(0, 0, 0)$. This choice of replacement pixels aligns with standard practice. For Deletion, we compute the probability of the label class after perturbing the pixels in descending order sorted by attribution values. In the end, we obtain a curve of the predicted probability against the ratio of perturbed pixels. Then, we compute the area under the curve (AUC). A lower Deletion AUC indicates better

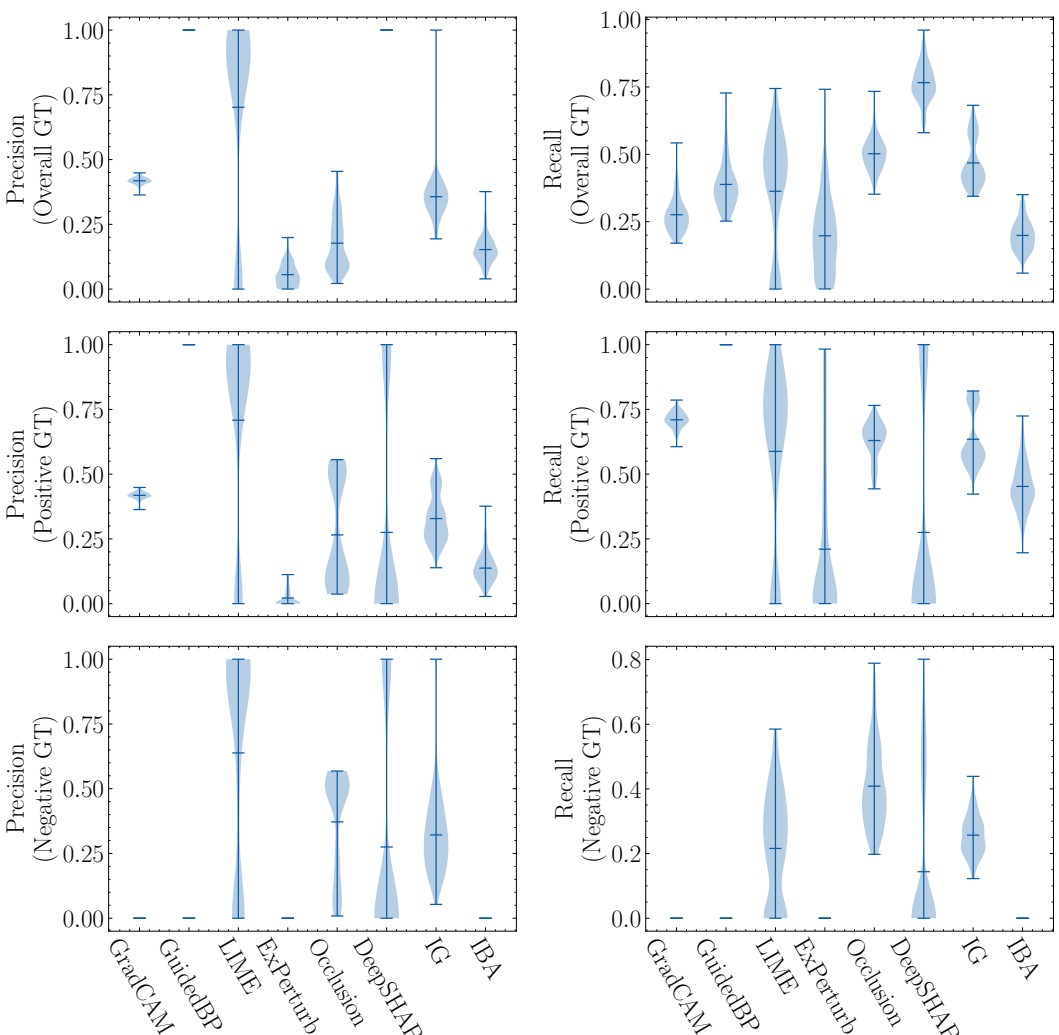

Figure 19: Ground-truth-based evaluation in the multi-color-sum setting. In each violin plot, the maximal, average, and minimal values are marked with horizontal bars. The first row. Note that only Occlusion, DeepSHAP, IG, and IG* yield negative attribution. For other attribution methods shown in the figure, we manually set their precision and recall associated with negative GT to 0.0.

performance of the attribution method. For Insertion, we compute the probability of the label class after inserting the pixels into a blank canvas with intensity $(0, 0, 0)$ in ascending order sorted by attribution values. Again, the predicted probability after insertion is recorded, allowing us to obtain a curve of the predicted probability against the ratio of inserted pixels. Then, we compute the AUC, and a higher Insertion AUC indicates better performance of the attribution method.

### F.4.2 EXPERIMENTAL RESULTS

As shown in Table 4 and Figure 8a, Insertion/Deletion display limited consistency with GT-based evaluation, when the Unseen Data Effect is present.

### F.5 SENSITIVITY-N

### F.5.1 EXPERIMENTAL SETUP

The implementation of Sensitivity-N in the multi-color-sum setting is consistent with the original paper. For a specific number of perturbed pixels $N$, we randomly select $N$ pixels in the image,

Table 4: Rankings of attribution methods in the multi-color-sum setting when evaluated with ground truth masks, Insertion/Deletion, and Sensitivity-N. Correlation denotes the Spearman's rank correlation with the $F_1$-score (computed using the positive ground truth masks).

| Attribution methods | Ranking | | | |
|---|---|---|---|---|
| | $F_1$-score | Insertion | Deletion | Sensitivity-N |
| GradCAM | 8 | 4 | 6 | 3 |
| GuidedBP | 3 | 5 | 4 | 1 |
| LIME | 1 | 3 | 3 | 2 |
| ExPerturb | 4 | 6 | 7 | 8 |
| Occlusion | 6 | 2 | 5 | 5 |
| DeepSHAP | 5 | 8 | 2 | 6 |
| IG | 7 | 1 | 1 | 4 |
| IBA | 2 | 7 | 8 | 7 |
| Correlation | − | 0.02 | 0.47 | 0.65 |

and perturb them to the default baseline $(0, 0, 0)$. After that, we feed the perturbed image into the neural network and obtain the predicted probability of the label class. Additionally, we record the sum of attribution of the perturbed pixels. This process is repeated for 100 times, yielding a pair of vectors with 100 elements, where the first vector represents the predicted probability in 100 repetitions and the second vector represents the sum of attribution. Next, we compute the Pearson Correlation Coefficient between the vectors of predicted probability and sum of attribution. With $N$ increasing from 1 to $224 \times 224$, we obtain a curve of Pearson Correlation Coefficient. The greater the correlation is, the better the attribution method performs.

### F.5.2 EXPERIMENTAL RESULTS

The Sensitivity-N curves are shown in Figure 8c. First, we observe that the performance of the attribution methods is much less distinguishable than in the single-color-modulo case. For instance, the curves of GradCAM, GuidedBP, and Occlusion intersect with one another multiple times, making it harder to conclude which method displays superior performance. Second, DeepSHAP produces a negative correlation curve. This can be interpreted by our previous analysis that DeepSHAP often misidentifies the sign of contributing pixels. Therefore, the sum of attribution of perturbed pixels is much less aligned with the predicted probability of the perturbed image. Third, in order to rank these methods and compare the ranking with the ranking on GT-based evaluation, we simply compute the average correlation over all $N$s. As demonstrated in Table 4, the Sensitivity-N evaluation result is much less correlated with the ground-truth-based evaluation compared to that in the single-color-modulo setting.

## G EXPERIMENTS IN THE MULTI-COLOR-SUM SETTING (WITHOUT UNSEEN DATA EFFECT)

In our default settings for the Accumulator module within the multi-color setting, we employ a non-uniform weight initialization. In this section, we present supplementary empirical results from the multi-color-sum setting, where we implement a uniform weight initialization in the Accumulator. Please note that: (1) the weight initialization schemes do not impact the Accumulator output, ensuring that the model still achieves $100\%$ accuracy on the synthetic dataset; and (2) the color detector retains redundant channels and consequently, remains susceptible to Unseen Data Effect. In the following subsections, the evaluation settings are identical as the Appendix F, except for the number of redundant channels in the Accumulator.

Based on the data provided in Table 5, it's apparent that the correlations between ground-truth-based evaluation and other metrics do not show significant improvement. This suggests that simplifying the Accumulator network by utilizing uniform weights does not necessarily mitigate the adverse behavior of Insertion/Deletion and Sensitivity metrics that stem from Unseen Data Effect.

Table 5: Rankings of attribution methods in the multi-color-sum setting (without Unseen Data Effect). Attribution methods are evaluated with ground truth masks, Insertion/Deletion, and Sensitivity-N. Correlation denotes the Spearman's rank correlation with the $F_1$-score (computed using the overall ground truth masks).

| Attribution methods | Ranking | | | |
|---|---|---|---|---|
| | $F_1$-score | Insertion | Deletion | Sensitivity-N |
| GradCAM | 5 | 5 | 6 | 6 |
| GuidedBP | 6 | 6 | 4 | 1 |
| LIME | 1 | 4 | 3 | 4 |
| ExPerturb | 7 | 8 | 7 | 8 |
| Occlusion | 2 | 3 | 5 | 5 |
| DeepSHAP | 3 | 1 | 2 | 2 |
| IG | 4 | 2 | 1 | 3 |
| IBA | 8 | 7 | 8 | 7 |
| Correlation | – | 0.42 | 0.61 | 0.81 |

## G.1 GROUND-TRUTH-BASED EVALUATION

Figure 20 illustrates the violin plots of various attribution methods, evaluated with different sets of GT.

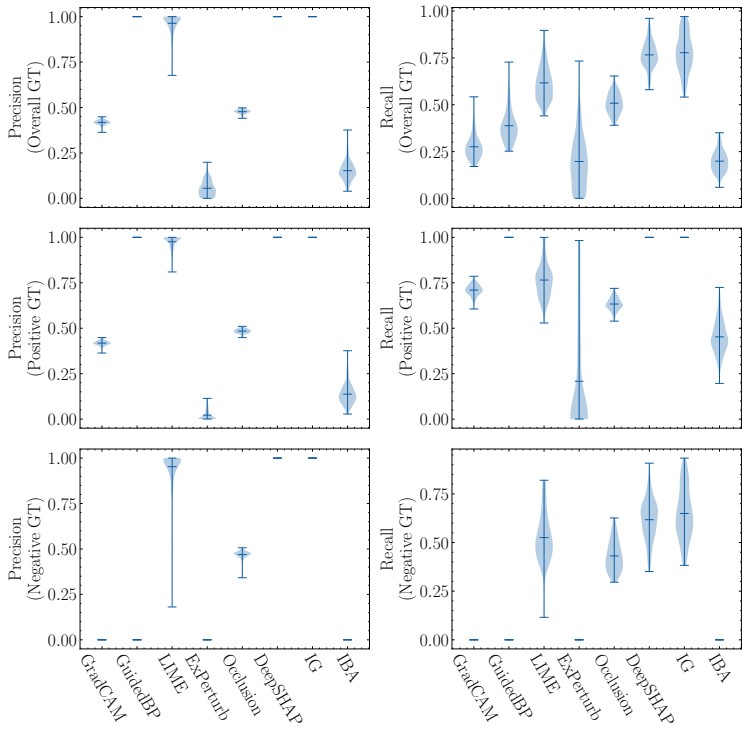

Figure 20: Ground-truth-based evaluation results in the multi-color-sum setting without Unseen Data Effect.

## G.2 INSERTION/DELETION

Figure 8b shows the Insertion curve.

### G.3 SENSITIVITY-N

Figure 8d depicts the correlation curves.

## H   BROADER IMPACTS

In this work, we introduce an evaluation framework designed specifically for feature attribution methods. A notable gap in the research community has been the absence of reliable evaluation metrics, leading to various complications. For example, numerous feature attribution methods have been proposed that, in some cases, even yield inconsistent attribution results given identical inputs. Our approach guarantees its validity for evaluation purposes because of the fully synthetic nature. Our synthetic settings may appear primitive compared to real-world datasets and neural networks trained on such datasets. However, they provide a controlled laboratory environment, enabling a thorough examination of feature attribution methods prior to deployment. This facilitates a robust evaluation and refinement process for attribution methods.

## I   ADDITIONAL STUDY ON SEGMENTATION METHODS OF LIME

In Section 5.2, we observed that the segmentation method significantly affects LIME's performance. This observation prompted us to explore a more advanced segmentation approach capable of effectively grouping features that function together and identifying independent feature groups. We conducted a comparative analysis of LIME attribution maps generated using Quickshift (Vedaldi & Soatto, 2008), Felzenszwalb (Felzenszwalb & Huttenlocher, 2004), and the recent segmentation approach Segment Anything Model (SAM) (Kirillov et al., 2023). Our comparison in the multi-color-sum setting, as shown in Figure 21, reveals a substantial increase in faithfulness when SAM is utilized as the segmentation method. Furthermore, we employ LIME for explaining two ImageNet-pretrained models: VGG (Simonyan & Zisserman, 2015) and ViT Dosovitskiy et al. (2020), respectively. Visual inspection of the attribution maps on VGG16 (Figure 22) and ViT-B-16 (Figure 23), indicates that SAM provides a prior of more cohesively grouped features and significantly reduces noise in the LIME attribution maps. These attribution maps are much better aligned with human perception as well. Moreover, this finding is consistent across the CNN and ViT architectures. This study underscores that the findings in AttributionLab can assist researchers to identify strategies for enhancing existing attribution methods, and it demonstrates that these enhancements can generalize to complex real-world scenarios.

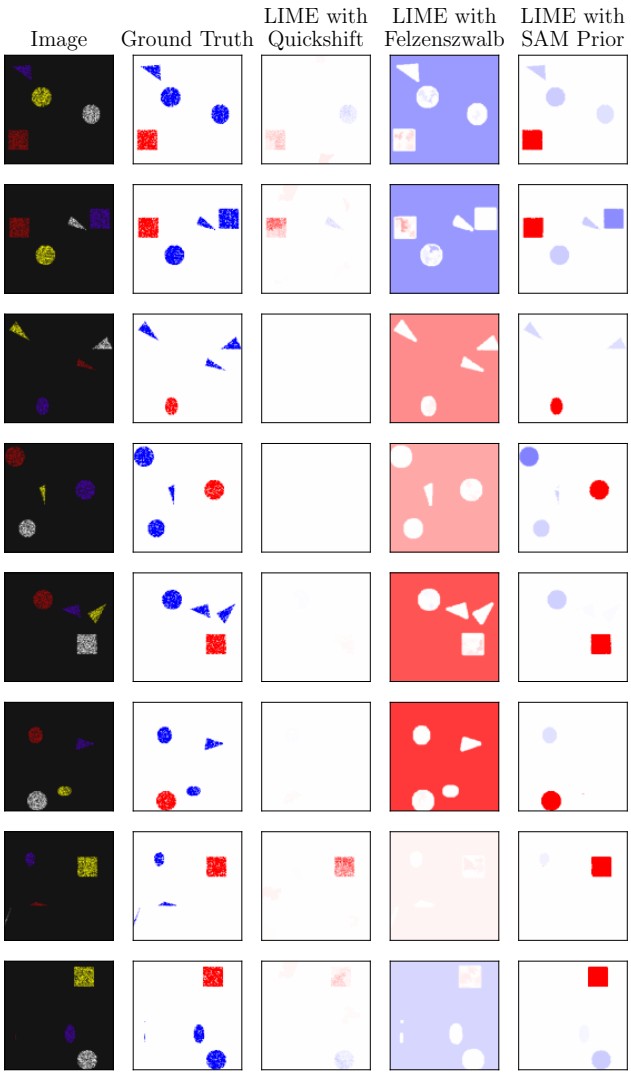

Figure 21: LIME in the multi-color-sum setting with different segmentation methods. In contrast to Quickshift (Vedaldi & Soatto, 2008) and Felzenszwalb (Felzenszwalb & Huttenlocher, 2004), SAM (Kirillov et al., 2023) offers markedly improved segmentation masks for inputs. This improvement significantly boosts the faithfulness of attribution maps, ensuring closer alignment with the ground truth attribution map.

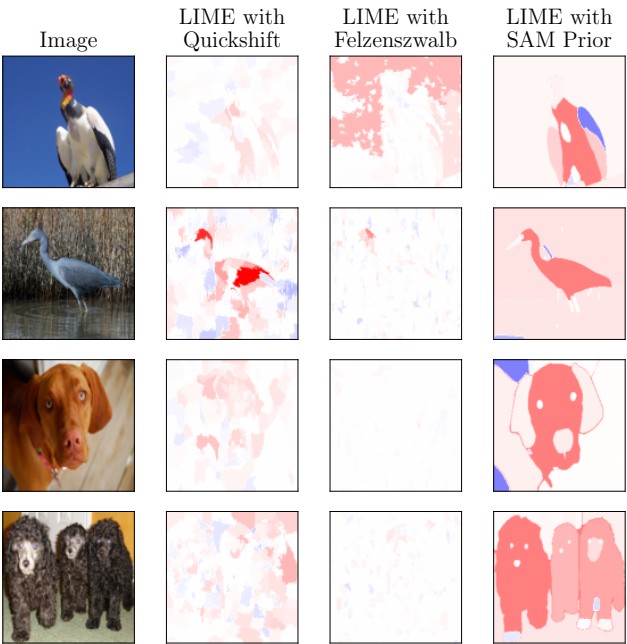

Figure 22: LIME attribution maps for VGG16 with different segmentation methods. In line with results in the multi-color-sum setting, SAM segments input features into more compact and semantically meaningful groups. Consequently, the LIME attribution maps generated using SAM exhibit reduced noise and are visually more congruent with human perception.

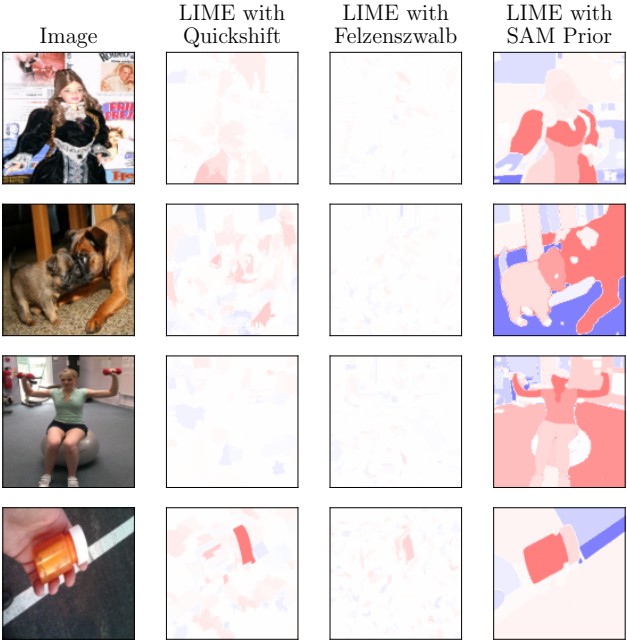

Figure 23: LIME attribution maps for ViT-B-16 with different segmentation methods. These attribution maps show a significant enhancement in attribution quality when LIME incorporates SAM, and this finding generalizes to vision transformers.

