# OpenReview forum: "AttributionLab: Faithfulness of Feature Attribution Under Controllable Environments"
_ICLR.cc/2024/Conference — Submitted to ICLR 2024_

### Official Review · Reviewer_MuCE · 2023-10-31

**Soundness:** 3 good
**Presentation:** 3 good
**Contribution:** 3 good
**Rating:** 6
**Confidence:** 4

**Summary:**

This paper proposes AttributionLab, a dataset and handcrafted model as a test bed for feature attribution methods. Since the data, both inputs and labels, and the model weights have all be specially crafted so that there ground truth feature importance is known, AttributionLab allows for testing feature importance methods in a controlled environment.

**Strengths:**

1. Clearly written -- the writing is easy to understand and the paper is well organized.
2. Well motivated -- assessing feature attribution faithfulness is extremely hard in practice.
3. Potentially useful -- if practitioners need faithfulness in their explainers, this benchmark may be a useful tool for comparing two explainers.

**Weaknesses:**

1. Faithfulness is a complicated question. The implication in this paper is that without AttributionLab one might not have ground truth feature importance scores. This paper argues that it is therefore hard to measure faithfulness, but leaves me wondering if there is value in defining the measure at all. Perhaps this weakness is really an issue in the field at large and not the responsibility of this paper.
2. The experiments are all visual. It is my understanding that feature attribution methods are often used in practice on tabular problems as well as language. If AttributionLab would benefit from including ways of evaluating XAI tools in those other very popular domains.

**Questions:**

1. The Disagreement Problem (Krishna et al. 2022) is cited but not discussed much. In light of this work, knowing that feature attribution methods tend to disagree, what level of faithfulness int he Attribution Lab do the authors ascribe to "sufficient" for use in practice? In other words -- we have existing work that shows that two methods won't find the same feature importance scores, so less than 100% faithful must be tolerable -- do the authors have intuition or recommendations for practitioners on faithfulness ranges that are good/bad?
2. With respect to IG and Occlusion, an interesting point is made on the importance of the baseline value for these methods. I understand how a 'good' choice is made in AttributionLab and how we can conclude that the baseline matters. Is there a method for picking real baselines on real data? On real tabular data? or natural language data?

---

> ### Author Response · Authors · 2023-11-18
> **Response to Reviewer MuCE**
>
> We are thankful for the reviewer's positive acknowledgment and constructive insights. Below, we delve into detailed responses to the concerns.
>
> ### W1: Definition of faithfulness
> The reviewer's point about the abstract nature of "faithfulness" in feature attribution is indeed thought-provoking. This concept varies in interpretation among different researchers, leading to a spectrum of understandings. As highlighted in our Related Work section, the existing formalizations of faithfulness in the literature come with their own set of issues. A common complication is that these definitions often intertwine with other intricate concepts, such as out-of-distribution (OOD) effects.  In contrast, using our framework, we can materialize the abstract notion of alignment with the model, because we know how the model uses each feature. This approach simplifies the assessment of faithfulness by clearly separating it from other confounding factors, including OOD effects.
>
> ### W2: Extension of AttributionLab to other data modalities and tasks
> We agree that extending to other modalities (language and table) is a natural extension of AttributionLab. However, even in the visual domain itself we had to push many experiments and results to the appendix section. Extension to tabular, graphs and language can be standalone works themselves. The design of AttributionLab can be extended to various data modalities and tasks. The essential requirement for researchers is to manually craft both the model and the data in a manner that ensures the model’s predictive behavior is completely transparent. This fundamental principle remains consistent across different modalities and tasks, indicating no theoretical impediments to extending AttributionLab's application.
>
>
> ### Q1: Faithfulness thresholds for a method
> We appreciate the reviewer for raising this point. It is important to note that our AttributionLab is primarily intended as a sanity check or debugging tool, rather than a benchmarking platform. Consequently, if an attribution method exhibits less than 100% faithfulness in our tests, it implies that the method may have inherent limitations. However, this does not automatically render the attribution method unsuitable for real-world applications. As we show in the paper, we can see how and why the attribution is not faithful to the model. For instance, our observations with DeepSHAP indicate that while it accurately attributes to the contributing features, it fails to distinguish between positively and negatively contributing features effectively.
>
> ### Q2: How to choose a proper baseline
> Thank you for bringing this to our attention. Indeed, our observations emphasize the significant impact of baseline choice on the effectiveness of attribution methods, particularly when the baseline is known. Our work underscores the crucial link between the choice of baseline and the model's learned distribution. To select an appropriate baseline, one must have prior knowledge of the model's learned distribution, which AttributionLab facilitates due to its designed model and data.
>
> However, in many existing studies, the baseline is chosen based solely on the data. Common practices range from using simple black pixels (value 0) to average pixel values. Some approaches have employed more sophisticated techniques, such as inpainting the input when an object is removed, to predict the baseline for the excised portion. But these methods primarily consider only the data. Consider a model trained on CIFAR-10, which is not encoded to recognize a broader range of natural images beyond its 10 classes. If you input an image of a car on a dirt road into both a CLIP model and a CIFAR-trained model, the appropriate baselines would differ. For the CLIP model, using the dirt value as a baseline might be more appropriate, as it's likely been trained on images of both dirt and cars. However, for the CIFAR-trained model, the dirt represents an OOD value. Thus, a proper baseline should be within the distribution that the model has learned, which in itself is a challenging problem.
>
> Within AttributionLab, we can accurately determine baseline values because we fully understand the model's internal computation process. In real-world scenarios, we can improve the likelihood of choosing the right baselines by considering the specific task at hand. For example, for a model trained on chest X-rays, a good baseline might be the background values of chest X-rays (close to black), as the model is extensively trained on images characterized by stark black-and-white contrast.

---

> ### Author Response · Authors · 2023-11-20
> **A Kind Reminder: Review Response Awaiting Your Valued Feedback**
>
> Dear Reviewer,
>
> As the deadline for the discussion nears, we would like to gently ask you to join. We think we have addressed your concerns and we look forward to further discussions and guidance.
>
> Best regards,
>
> Authors of AttributionLab

---

> > ### Comment · Reviewer_MuCE · 2023-11-21
> > **Thank you for the responses**
> >
> > I appreciate the authors' explanations to all of my concerns. Mostly, these discussions answer my questions, but I think the paper would be significantly stronger with these discussions and extensions included. I will keep my score and I advocate that this paper be accepted.

---

> > > ### Author Response · Authors · 2023-11-23
> > > **Thank you for your feedback and explicit advocation for acceptance**
> > >
> > > Thank you for raising these discussion points. We will absolutely incorporate these discussions as we believe they truly strengthen the message of our work. Thank you for your support.

---

### Official Review · Reviewer_gSud · 2023-11-01

**Soundness:** 3 good
**Presentation:** 3 good
**Contribution:** 3 good
**Rating:** 5
**Confidence:** 2

**Summary:**

The paper sets up a controlled environment to evaluate feature attribution algorithms (i.e., algorithms to determine which input features affect the network's output) for neural networks. To do this both the data is constructed in a way that each input pixel affects the network's output and the network weights are specifically designed (not trained) to behave in a specific manner, such that for a given input it is known a priori which weights and which inputs should affect the output in which way. The evaluation of several popular attribution mechanisms shows differences of algorithms in their faithfulness and reaction to unseen data.

**Strengths:**

Evaluating attibution methods for their faithfulness and accuracy is important, especially for fields such as explainable AI. A strictly controlled environment and network to evaluate these approaches therefore makes sense.
The setup of the environment and of the neural net seems to make sense and allows for controlled evaluations of different settings.

**Weaknesses:**

Overall it's unclear for me what the concrete message of the paper is, except that different attribution algorithms behave differently. What exactly can we learn from these experiments? Are their specific weaknesses of some of the methods? Should they only be used in specific circumstances? Do they need to be interpreted differently? Are some methods strictly better than others? Can the attribution algorithms be somehow improved based on the findings here?

**Questions:**

Basically, my main question is how the findings of this controlled study translate to the real world where the datasets and models are much more complicated? Do we need to change how the current attribution mechanisms are used or interpreted? Should we stop using some of them or change their hyperparameters?

---

> ### Author Response · Authors · 2023-11-18
> **Response to Reviewer gSud**
>
> We thank the reviewer for the valuable comments and observations. We have carefully considered and responded to each point in the following discussion.
>
> ### W1: Central messages of our work
> We appreciate the reviewer for bringing up this important aspect. We completely agree that merely illustrating the shortcomings of methods may not suffice, despite its insightful nature. In reality, our work is already in harmony with the concerns you highlighted. Unlike previous evaluations that primarily concentrate on benchmarking attribution methods, our approach involves framing the work as a debugging setup. Here, we systematically assess each attribution method independently, uncovering its flaws, and suggesting viable remedies. This distinctive approach underscores our commitment to addressing concerns constructively rather than just highlighting issues.
>
> **Stepping beyond benchmarking towards debugging setup**
> AttributionLab serves as a strong sanity check and debugging tool for testing feature attribution methods. By evaluating their alignment with ground truth attributions, we can test whether a method produces faithful attributions. In such a fully controllable environment, we can uncover the issues with attribution methods. For instance, we reveal in AttributionLab that IG is very sensitive to the baseline value, and IG performs substantially better when it uses the true baseline (for a complete discussion on IG and baseline, please check our response to reviewer MuCE, Q2). Additionally, our findings in AttributionLab have led to improvements in existing methods. For instance, we observed that LIME's performance is heavily dependent on the segmentation method. In our latest experiments, detailed in Figures 21 to 23 in Appendix I, we substantially enhanced LIME's faithfulness by employing a more accurate segmentation technique. Thus, AttributionLab not only aids researchers in uncovering the limitations of attribution methods but also directs them towards effective solutions for these challenges.
>
> We warmly invite the reviewer to participate further in this discussion. Your insights and suggestions are valuable in enhancing AttributionLab and advancing research in feature attribution.
>
> ### Q1: Generalization of the findings and potential improvements for attribution methods
>
> In our experiments conducted using AttributionLab, we extended our testing to real-world models, such as VGG—a representative CNN architecture—on large-scale datasets like ImageNet. These tests corroborated our initial findings from AttributionLab, demonstrating that the issues identified with attribution methods are indeed prevalent in real-world scenarios.
>
> The insights gained from AttributionLab have paved the way for enhancements in the attribution methods we tested. For example, our research indicates that DeepSHAP excels at localizing predictive features, albeit without accounting for the attribution sign. In scenarios where the goal is merely to identify features relevant to predictions, one could utilize the absolute values of DeepSHAP attributions. This approach is validated by the high Overall Precision and Overall Recall of DeepSHAP, as depicted in Figure 5(a). Additionally, our findings highlight the significant impact of segmentation methods on LIME attribution results. Leveraging more advanced segmentation techniques, such as the Segment Anything Model (SAM), could yield more semantic-aware segmentation masks for the subsequent LIME attribution. We observe that the attribution is faithful in terms of attribution sign as well. We have conducted additional experiments utilizing SAM in both AttributionLab and real-world environments. As detailed in Appendix I (Figure 21, Figure 22, and Figure 23), the LIME attribution maps generated with SAM demonstrate markedly improved faithfulness.  This enhancement is evident both in real-world scenarios and AttributionLab. Furthermore, this improvement of LIME attribution maps in real-world scenarios is consistent across both CNN and vision transformer (ViT) architectures. The LIME-SAM method seems to faithfully represent the contribution of different segments of the scene in terms of the scale of contribution and the sign of the contribution (positive/negative).

---

> > ### Comment · Reviewer_gSud · 2023-11-21
> > **Thank you.**
> >
> > Thank you for your response.
> > These details are indeed useful and can benefit the interpreation and useability of these methods. However, I think the paper might benefit from some work to more clearly incorporate those findings into the main part of the paper, ideally with some (practical) implications/advice about how your findings should be used in the future (such as your comments about better segmentation maps, etc).
> >
> > Based on this I'll keep my score since I think incorporating these things directly into the main paper will make the paper stronger. That said, I am not opposed to this paper being accepted here assuming the authors can integrate some of my feedback into a potential final version.

---

> > > ### Author Response · Authors · 2023-11-21
> > >
> > > Dear Reviewer gSud,
> > >
> > > We greatly appreciate your acknowledgment of our paper's contribution. We would like to clarify that the details highlighted in our last response are, in fact, already presented within the original submitted draft. This includes all aspects except for the new LIME+SAM experiment, which is a recent addition. We will revise our manuscript to make these aspects more explicit in our writing.
> > >
> > > Given that our work implicitly aligns with your concerns and requires clarifications only, we hope that this may encourage a reconsideration of your review score. Nevertheless, we highly appreciate your feedback and engagement and are already grateful.
> > >
> > > Warm regards,
> > >
> > > Authors of AttributionLab

---

> ### Author Response · Authors · 2023-11-20
> **A Kind Reminder: Review Response Awaiting Your Valued Feedback**
>
> Dear Reviewer,
>
> As the deadline for the discussion nears, we would like to gently ask you to join. We think we have addressed your concerns and we look forward to further discussions and guidance.
>
> Best regards,
>
> Authors of AttributionLab

---

### Official Review · Reviewer_2xtR · 2023-11-01

**Soundness:** 3 good
**Presentation:** 3 good
**Contribution:** 3 good
**Rating:** 6
**Confidence:** 2

**Summary:**

The paper proposes a new simulated environment for attribution faithfulness measurement. Attribution faithfulness refers to checking how well the attribution maps from different attribution methods such as Lime or integrated gradients align with the ground truth attribution map. This faithfulness can guide us to select which attribution method to rely more on. Existing methods of attribution faithfulness design a synthetic dataset in which the expected attribution maps are known and then after the network's training on this dataset, they check how well aligned are the predicted attribution maps with the ground truth attribution map. The authors argue that there is a fundamental flaw with this, because the neural network might not be actually using the ground truth attributions to perform the task. For instance, there might be some spurious features in the dataset which can lead to 100% train accuracy, and thus any attribution method that generates these spurious features in the predicted attribution map will be scored lower in faithfulness as the the predicted attribution (even though correct) will be different than ground truth attribution map.

Thus the authors argue that instead of just designing the dataset, any controlled environment should also design the neural network so that we exactly know which features are actually being used by the model. The authors design a dataset where the aim is to predict the dominant colour (in terms of number of pixels) and design the neural network in a way that it first detects the colour and then counts the number of pixels for that colours. The authors then use different attribution methods to predict the attribution maps for different images and then measure their faithfulness.

**Strengths:**

I really like the idea motivated in the paper about having control over not only the dataset but also over the design of the learning process to properly measure attribution faithfulness. The authors can also refer to other datasets which talk about spurious features in the dataset to motivate their reasoning.
Such a work will help model developers in the future to be able to better decide on proper attribution methods.

**Weaknesses:**

1. The authors mention/deisgn only one dataset and their learning process. Is it possible to also do this analysis over another synthetic dataset an show that the conclusions drawn from faithfulness are similar across the two datasets/learning processes?

**Questions:**

I have already mentioned it in the weakness section

---

> ### Author Response · Authors · 2023-11-18
> **Response to Reviewer 2xtR**
>
> We are thankful for the reviewer's positive acknowledgment and constructive insights. Below, we delve into detailed responses to the raised concerns.
>
> ### W1 & Q1: Extension to other tasks
> We sincerely thank the reviewer for acknowledging the contributions of our research. As detailed in Appendix D and Appendix E, we have an additional synthetic setting, termed single-color-modulo. This setting diverges from the multi-color-sum scenario by focusing on a regression task, rather than classification. We have made similar observations in the single-color-modulo setting as in the multi-color-sum setting and ImageNet. Moreover, the potential of our synthetic setting to be adapted for more complex models and tasks opens avenues for exploring feature attribution in greater depth. Throughout our paper, we present qualitative experiments demonstrating that our observations are also applicable to real-world scenarios, such as ImageNet-VGG. During the discussion phase, we introduced an enhanced variation of the existing LIME method, as depicted in Figures 21 to 23 in Appendix I. We also qualitatively demonstrate that this improvement effectively transfers to Vision Transformers (ViTs). We kindly invite the reviewer to engage further in this discussion, especially considering the promising potential of our work.

---

> > ### Comment · Reviewer_2xtR · 2023-11-22
> > **Thanks for your response.**
> >
> > Thanks to the authors for pointing the appendix sections D and E with additional synthetic setting experiments. I plan to keep my current rating as it is.

---

> ### Author Response · Authors · 2023-11-20
> **A Kind Reminder: Review Response Awaiting Your Valued Feedback**
>
> Dear Reviewer,
>
> As the deadline for the discussion nears, we would like to gently ask you to join. We think we have addressed your concerns and we look forward to further discussions and guidance.
>
> Best regards,
>
> Authors of AttributionLab

---

### Official Review · Reviewer_7XTH · 2023-11-05

**Soundness:** 3 good
**Presentation:** 3 good
**Contribution:** 2 fair
**Rating:** 6
**Confidence:** 4

**Summary:**

The paper proposes a synthetic based framework that can serve as a sanity check for attribution methods. The goal of the proposed framework is to test the faithfulness of the attribution methods that represents the similarity between attributed features and the true features used by model for prediction. Since the trained model could rely on unwanted or spurious feature for its prediction in order to effectively evaluate the attribution method, there is a need to have models that relies on known and reliable features. The proposed synthetic framework achieves this by defining custom tasks (for e.g dominant shape’s color prediction) - as the ground truth attribution is known as well as by manually setting the weights of models (CNN based classifier) that are the true solution for the given custom tasks. The paper hypothesizes that if the attribution methods output doesn’t align with the known GT attribution of the task then the attribution would be unreliable. The paper empirically evaluates different attribution methods and also proposes some improvement for those attribution methods.

**Strengths:**

The paper explores a synthetic attribution evaluation framework that has a library of predefined tasks (for eg. identify the dominant color in terms of number of pixels in grayscale and RGB input setting) along with the model’s weight that perfectly solves those custom tasks. The proposed framework shows evaluation of various existing attribution methods and proposes insights on how to improve some of these attribution methods. The paper provides detailed ablation experiments to analyze/evaluate the various aspects of attribution methods (positive/negative attribution features). Ablation experiments regarding the unseen data effects sheds light on the benefit of utilizing this synthetic framework to analyze the attribution approaches when out of distribution input are used for inference. The paper is well-written, highly organized and is easy to follow. The paper provides code for reproducibility and has readme instructions to use the codebase.

**Weaknesses:**

It would be helpful for a reader to get a better understanding of the following:

It would be interesting to see the framework evaluation on a larger scale that supports different architectures, tasks and problem domains. In order to guarantee or even to be in a good standing empirically, it could require more controllable environments (tasks along with the perfect model weights) to verify the faithfulness of an attribution method. Specifically, it would be interesting to see the following:
1. Framework that has support for transformer based architecture as the shift of computer vision models (and other domain problems) from CNN to transformer is rampant.
2. How would the evaluation of transformer based model look like, that already has a build-in support for attribution feature via visualizing the attention maps rather than relying on post-hoc attribution methods (this implicit attention feature attribution indeed should be accurate as this represents the inner computation graph itself).
3. Extending the library of tasks from simple classification setting (with underlying counting or simple arithmetic jobs) to more abstract setting that necessarily doesn’t satisfy the “Proposition 2 (Symmetry property)” (i.e. The addition/removal of any ground-truth pixel to/from the background equally affects the output of the designed neural network.)
4. To ensure that the verification of attribution method on these custom designed task with small network architecture (2 layer CNN with ReLUs) would generalize to attribution to a large model (that is generally used, with millions or billions of parameters) for the same attribution methods. This might require more complex benchmark tasks.

**Questions:**

It would be helpful if the paper can answer/comment the following questions/suggestions:

1. [question] In the paper for equation 3, is there a typo and should this be this instead ? “R(r, g, b) = 1 if Ci(r, g, b) = 1” -> “R(r, g, b) = 0 if Ci(r, g, b) = 1”
2. [suggestion] While stating “An attribution may seem reasonable to us, but the neural network may use other input features. Conversely, an attribution may seem unreasonable but be faithful and indeed reflect the features relevant to the neural networks.”, it might be helpful to cite “bug as features” paper [1] that supports this claim.
3. [suggestion] To further stress on the need of having robust model that doesn’t rely on spurious patterns to make predictions and properly evaluating the attribution methods (i.e. a setting where we know the GT attribution and have access to perfect model that fully solves this problems), it might be helpful to point this in related section by citing methods [2,3] that prove the fragility of trained models leads to attribution features that could be manipulated.

references:
[1] Adversarial Examples Are Not Bugs, They Are Features, NeurIPS 2019
[2] Robust Attribution Regularization, NeurIPS 2019
[3] Attributional Robustness Training using Input-Gradient Spatial Alignment, ECCV 2020

---

> ### Author Response · Authors · 2023-11-18
> **Response to Reviewer 7XTH (1/2)**
>
> We appreciate the reviewer's insightful feedback and take this opportunity to thoroughly address the concerns in the subsequent sections.
>
> ### W1: Support for transformers
> **Attribution is still not solved for CNNs and MLPs** The focus of the literature has primarily been on addressing the attribution problem within CNNs and MLPs, a challenge that continues to be unresolved. Furthermore, MLPs and CNNs are still widely used in recent large-scale models. For instance, MLPs are used for encoding Key, Query, and Value in transformers, while CNNs are used interspersed with attention in the UNet in Stable Diffusion. Therefore, feature attribution for CNNs and MLPs is still a very critical research topic. In our work, we are pleased to present a significant advancement towards a solution, utilizing a fully controllable environment for the first time. This novel approach not only serves as a strong sanity check and debugging tool for various methods but also exposes the unique properties and limitations of all existing methods we have tested.
>
> **It is feasible to extend the framework to Transformers** We showcased several CNN and MLP modules designed for various tasks. The versatility of our method extends to other setups such as regression and multimodal tasks. This extendibility is a key strength of our approach. For future work we can propose transformers for the proposed tasks, as there is nothing fundamentally different (in fact MLPs are part of the equation). While exploring the application of existing attribution methods to transformers remains an intriguing prospect, our current focus on CNNs and MLPs has enabled us to identify specific issues and properties of these models. As most of these attribution methods themselves can be also applied to transformers. For instance in a new experiment (shown in Figure 21, 22, 23 in Appendix I), we show that an extension of LIME which was found to work well in AttributionLab, seems promising in ViT as well.
>
> ### W2:  Attention in transformers implicitly represents attribution
>
> **Attention being explanation is disputed** Attention maps offer insights into the interactions between input tokens in transformers. However, their effectiveness as reliable explanations has been a topic of considerable debate. These attention maps visualize attribution between Query (Q) and Key (K). However, Q and K are outcomes of encoding through two MLPs in the attention layer, which means that attention maps offer only a partial view of the entire attention process, as the mappings to Q and K remain unknown. Additionally, the complexity introduced by multiple layers and attention heads further complicates the interpretation of this information. Besides, previous studies also question the validity of attention maps being explanations. Jain et al. [1] and Serrano et al. [2] have highlighted inconsistencies between attention maps and gradient-based or occlusion-based methods. Furthermore, Attanasio et al. [3] reveal in a benchmark that attention maps are only partially faithful for transformers. These studies collectively suggest that attention maps may not fully capture the nuances of contributions of Q and K. In light of these insights, we primarily focus on studying the post-hoc attribution methods in this work.
>
> **Attribution is still needed for transformers** Feature attribution methods explain models from different aspects. One of the mainly accepted ones in the concept of contribution (credit assignment). This concept assigns contribution to features based on their effect on the output when they are removed from the input. Most attribution methods are based on this aspect. This aspect is orthogonal to attention and activation values. Thus even if attention values were understood properly, they would still shed light on one facet of transformers and the contribution assignment methods still reveal different insights.
>
> [1] Jain, S., & Wallace, B.C. (2019). Attention is not Explanation. North American Chapter of the Association for Computational Linguistics.
>
> [2] Bai, B., Liang, J., Zhang, G., Li, H., Bai, K., & Wang, F. (2021, August). Why attentions may not be interpretable?. In Proceedings of the 27th ACM SIGKDD Conference on Knowledge Discovery & Data Mining (pp. 25-34).
>
> [3] Attanasio, G., Nozza, D., Pastor, E., & Hovy, D. (2022). Benchmarking post-hoc interpretability approaches for transformer-based misogyny detection. In Proceedings of NLP Power! The First Workshop on Efficient Benchmarking in NLP. Association for Computational Linguistics.

---

> ### Author Response · Authors · 2023-11-18
> **Response to Reviewer 7XTH (2/2)**
>
> ### W3: Extension to other tasks
> **Other tasks already exist**
> In our study, we have expanded our synthetic setting to include a regression task called single-color-modulo, as detailed in Appendix D and E. Adhering to the core principles of AttributionLab, we manually crafted both the MLP-based neural network and the corresponding dataset.
>
> **Propositions and axioms of feature attribution** We intentionally enforced the symmetry and sensitivity propositions in both the data and model design. These properties stem from the sensitivity and symmetry axioms [4] of feature attribution. Incorporating them into the designs gives us the ability to see if different attribution methods are aligned in practice with these axioms. These axioms outline desired properties for attribution methods. However, it does not mean we need to be bound by these properties. Depending on the properties of the attribution method we aim to evaluate, we can adapt our setups to include more, fewer, or different axioms.
>
> **Symmetry, Sensitivity are not limits, but additions** These properties facilitate the evaluation of corresponding properties in attribution methods, as previously discussed. Creating setups that lack these properties is relatively straightforward. For example, consider a simplified color classifier setup that determines the presence of a color. Imagine an input with two blue shapes and one yellow shape. In this scenario, the setup lacks sensitivity, as the classification can be triggered by just one colored pixel, and it's not inherently clear which specific pixel leads to the classification. The absence of these properties restricts our evaluation capabilities. Therefore, incorporating properties like sensitivity and symmetry adds complexity to the setup but significantly enhances its evaluative potential, a benefit that extends to more abstract and complex tasks. Hence, including the sensitivity or symmetry axiom is not a limitation or drawback. Rather, it's an addition to evaluate these properties within attribution methods.
>
>
> [4] ​​​​​​​​​​Sundararajan, M., & Najmi, A. (2020, November). The many Shapley values for model explanation. In International conference on machine learning (pp. 9269-9278). PMLR.
>
> ### W4: Generalization to larger scales
>
> *Minor remark*: We have several CNN and MLP modules, and the color detector alone is 4 layer CNN (The models are ~ 10 layers). The tasks range from modulo computation to counting and color detection.
>
> We agree that the absence of a formal proof for generalization, as you highlighted, represents a limitation within the ground truth-based approach, as also acknowledged in our paper. Despite this, the research on interpretability and the feature attribution problem at a large scale remains inconclusive and open. We encounter the disagreement problem [5] where although the methods are theoretically sound, they diverge in their outcomes. Our paper contends that the exploration of a controlled setup in this research direction is opening new opportunities to address this issue. Please consider the following:
>
> - **Sanity check and debugging** The framework functions as a sanity check and provides an environment for debugging methods. While it may not offer the ultimate solution within this paradigm, it allows us to identify what doesn't work. This is akin to the strategy employed in the sanity checks [6] paper, which critically evaluated various attribution methods prevalent in prior research. In addition, our framework enables us to understand how certain approaches fail and suggests potential solutions to address the issues.
>
> - **Toy models are also used for understanding large models (e.g. LLMs)** With the current explosion in scale, the problem gets more and more convoluted to interpret these large foundation models. In order to understand them, the recent mechanistic interpretability works [7] also use toy transformers to understand their behavior.
>
> - **No method works at current small scale** Despite the simplicity of the tasks, none of the attributions produces perfect attributions. In an ideal scenario, an attribution should perform flawlessly in AttributionLab (or any controlled environment).
>
>
> [5] Krishna, S., Han, T., Gu, A., Pombra, J., Jabbari, S., Wu, S., & Lakkaraju, H. (2022). The disagreement problem in explainable machine learning: A practitioner's perspective. arXiv preprint arXiv:2202.01602.
>
> [6] Adebayo, J., Gilmer, J., Muelly, M., Goodfellow, I., Hardt, M., & Kim, B. (2018). Sanity checks for saliency maps. Advances in neural information processing systems, 31.
>
> [7] Olsson, et al., "In-context Learning and Induction Heads", Transformer Circuits Thread, 2022.
>
> ### Q1, Q2, and Q3
> We thank the reviewer for pointing out the typo and suggesting additional related work, and we have updated the draft and incorporated these suggestions.

---

> ### Author Response · Authors · 2023-11-20
> **A Kind Reminder: Review Response Awaiting Your Valued Feedback**
>
> Dear Reviewer,
>
> As the deadline for the discussion nears, we would like to gently ask you to join. We think we have addressed your concerns and we look forward to further discussions and guidance.
>
> Best regards,
>
> Authors of AttributionLab

---

### Meta-Review · Area_Chair_mwdi · 2023-12-10

**Metareview:**

The paper proposes a new simulated/controlled environment for evaluating attribution faithfulness. Attribution faithfulness refers to checking how well the attribution maps from different attribution methods such as Lime or Integrated Gradients align with the ground truth attribution map. It can then be used as a sanity check for attribution methods. To that end, in the paper both the data is constructed in a way that each input pixel affects the network's output and the network weights are specifically designed to behave in a specific manner, such that for a given input it is known a priori which weights and which inputs should affect the output in which way. When evaluating several popular attribution mechanisms with this framework, it highlights differences in algorithms' faithfulness and their reactions to unseen data. It then proposes improvement for those attribution methods.

All reviewers think it is an interesting idea to set up an evaluation framework for attribution methods, but the conclusion from this evaluation framework is unclear. And it is questionable whether this method could generalize to different data and tasks, or scale to larger networks.

**Justification For Why Not Higher Score:**

All reviewers think it is an interesting idea to set up an evaluation framework for attribution methods, but the conclusion from this evaluation framework is unclear. And it is questionable whether this method could generalize to different data and tasks, or scale to larger networks.

**Justification For Why Not Lower Score:**

N/A

---

### Decision · Program_Chairs · 2024-01-16

Reject